# REFLECTIVE POLICY OPTIMIZATION

## ABSTRACT

On-policy reinforcement learning methods, such as Trust Region Policy Optimization (TRPO) and Proximal Policy Optimization (PPO), often require significant data to be collected at each update, giving rise to issues of sample inefficiency. This paper introduces a novel extension to on-policy methods called Reflective Policy Optimization (RPO). RPO's fundamental objective is amalgamating prior and subsequent state and action information from trajectory data to optimize the current policy. This approach empowers the agent to engage in introspection and introduce modifications to its actions within the current state to a certain degree. Furthermore, theoretical analyses substantiate that our proposed method not only upholds the crucial property of monotonically improving policy performance but also adeptly contracts the solution space of the optimized policy, consequently expediting the training procedure. We empirically demonstrate the feasibility and efficacy of our approach in reinforcement learning benchmarks, culminating in superior performance in terms of sample efficiency.

## 1    INTRODUCTION

On-policy reinforcement learning (RL) aims to learn an optimal mapping from states to actions based on performance criteria through the trajectory gained by interacting with the underlying environment in terms of performance criteria. Proximal Policy Optimization (PPO) (Schulman et al., 2017) is one of the most typical of these algorithms due to its simplicity and effectiveness and has been successfully applied in many domains, such as Atari games (Mnih et al., 2015), continuous control tasks (Dhariwal et al., 2017), and robot control (Lillicrap et al., 2016). However, existing algorithms optimize the policy by a state-action pair and don't directly consider the impact of this trajectory's subsequent states and actions, which may be a reason for sample inefficiency.

In previous studies (Mnih et al., 2015; van Hasselt et al., 2016; Schulman et al., 2015; 2017; Haarnoja et al., 2018; Silver et al., 2014; Fujimoto et al., 2018), basically the current policy is optimized using the value function of the current state. An open question: the value function potentially contains information about the subsequent data, is it the best way to optimize a policy using only value functions? The answer is definitely not. Let's start with an example. Considering an environment with a "cliff", what would an agent do if it performed an action under a state and fell into a "cliff"? This action is dangerous, so the agent will avoid performing it. Meanwhile, this state might also be hazardous because in the next time the agent reaches this state again, and it is likely to perform the same action. Hence, the agent must also avoid returning to this state and keeping out of this state as much as possible. The previous action when reaching this state also needs to directly avoid being performed, duo to the fact that it is possible to fall into that state again. The same result is found for the "treasure" environment. Subsequent data can convey positive and negative information to the previous states and actions. Hence, it is necessary to optimize the previous action directly with the subsequent state-action pairs information, not only through the value function. Intuitively, the direct use of the subsequent data may speed up the convergence of the algorithm and improve sample efficiency. For all we know, most existing algorithms lack this power, which directly exploits the relationship between the pair of trajectory data to optimize the policy. We illustrate that our proposed method has this ability by a toy example in the experimental section.

In this paper, we propose a simple on-policy algorithm that directly optimizes the policy by combining the relationship between the trajectories' previous and next state-action pairs. In other words, the proposed method considers the current state-action pair and the effect of the subsequent pair of trajectories. In this way, the optimized policy can be reflective. The proposed algorithm is called a

reflective policy optimization algorithm (RPO). The method proposed in this paper is fundamentally different from multi-step reinforcement learning methods (De Asis et al., 2018; Duan & Wainwright, 2023; Hernandez-Garcia & Sutton, 2019). Multi-step algorithms such as TD($\lambda$) (Sutton & Barto, 1998) work on the value function evaluation and are not directly involved in policy optimization. Although better results are produced in this way (Hessel et al., 2018), their theory is insufficient, limiting their application. The proposed method in this paper directly employs the previous and next information of trajectories on policy optimization, and we give a nice theory. We present a novel policy improvement lower bound. We show that in addition to satisfying the desirable property of the monotonic improvement of policy performance, our proposed method can effectively reduce the solution space of the optimized policy, speeding up the algorithm's training procedure. Our proposed method is combined with the PPO computational framework (Schulman et al., 2017) to present a practical version. Finally, we verify the feasibility and effectiveness of the proposed method by a toy example and achieve better performance on RL benchmarks (Brockman et al., 2016).

## 2 PRELIMINARIES

### 2.1 MARKOV DECISION PROCESS

Commonly, the reinforcement learning problem can be modeled as a Markov Decision Process (MDP), which is described by the tuple $\langle \mathcal{S}, \mathcal{A}, P, R, \gamma \rangle$ (Sutton & Barto, 1998). $\mathcal{S}$ and $\mathcal{A}$ are the state space and action space respectively. The function $P(s'|s, a) : \mathcal{S} \times \mathcal{A} \times \mathcal{S} \longmapsto [0, 1]$ is the transition probability function from state $s$ to state $s'$ under action $a$. The function $R(s, a) : \mathcal{S} \times \mathcal{A} \longmapsto \mathbb{R}$ is the reward function. And $\gamma \in [0, 1)$ is the discount factor for long-horizon returns. In a state $s$, the agent performs an action $a$ according to a stochastic policy $\pi : \mathcal{S} \times \mathcal{A} \longmapsto [0, 1]$ (satisfies $\sum_a \pi(a|s) = 1$). The environment returns a reward $R(s, a)$ and a new state $s'$ according to the transition function $P(s'|s, a)$. The agent interacts with the MDP to give a trajectory $\tau$ of states , actions, and rewards: $s_0, a_0, R(s_0, a_0), \cdots, s_t, a_t, R(s_t, a_t), \cdots$ over $\mathcal{S} \times \mathcal{A} \times \mathbb{R}$ (Silver et al., 2014). Under a given policy $\pi$, the state-action value function and state-value function are defined as

$$Q^\pi(s_t, a_t) = \mathbb{E}_{\tau \sim \pi}[G_t | s_t, a_t],$$
$$V^\pi(s_t) = \mathbb{E}_{\tau \sim \pi}[G_t | s_t],$$

where $G_t = \sum_{i=0}^\infty \gamma^i R_{t+i}$ is the discount return, and $R_t = R(s_t, a_t)$.

It is clear that $V^\pi(s_t) = \mathbb{E}_{a_t} Q^\pi(s_t, a_t)$. Correspondingly, advantage function can be represented $A^\pi(s, a) = Q^\pi(s, a) - V^\pi(s)$. We know that $\sum_a \pi(a|s) A^\pi(s, a) = 0$.

Let $\rho^\pi$ be a normalized discount state visitation distribution, defined

$$\rho^\pi(s) = (1 - \gamma) \sum_{t=0}^\infty \gamma^t \mathbb{P}(s_t = s | \rho_0, \pi),$$

where $\rho_0$ is the initial state distribution (Kakade & Langford, 2002). Similarly, $\rho^\pi(\cdot|s, a)$ can be defined and denotes the conditional visitation distribution under state $s$ and action $a$. And the normalized discount state-action visitation distribution can be represented $\rho^\pi(s, a) = \rho^\pi(s)\pi(a|s)$. We make it clear from the context whether $\rho^\pi$ refers to the state or state-action distribution.

The goal is to learn a policy that maximizes the expected total discounted reward $\eta(\pi)$, defined

$$\eta(\pi) = \mathbb{E}_{\tau \sim \pi} \left[ \sum_{i=0}^\infty \gamma^i R(s_{t+i}, a_{t+i}) \right].$$

The following identity indicates that the distance between the policy performance of $\pi$ and $\hat{\pi}$ is related to the advantage over $\pi$ (Kakade & Langford, 2002):

$$\eta(\pi) = \eta(\hat{\pi}) + \frac{1}{1 - \gamma} \mathbb{E}_{s, a \sim \rho^\pi} \left[ A^{\hat{\pi}}(s, a) \right]. \tag{1}$$

# 3  THE GENERALIZED SURROGATE FUNCTION

Some admirable algorithms obtain good properties by modifying the right-hand side of Eqn. (1), for example, Trust Region Policy Optimization (TRPO) algorithm (Schulman et al., 2015) optimizes the lower bound of policy improvement by replacing $\rho^\pi$ with $\rho^{\hat\pi}$ under state $s$, and offers better theoretical properties, i.e. monotonic improvement of policy improvement. Below, we give an equational relation between before and after replacements.

**Lemma 3.1.** *Consider a current policy $\hat\pi$, and any policies $\pi$, we have*

$$\mathbb{E}_{s,a\sim\rho^\pi}A^{\hat\pi}(s,a) - \mathbb{E}_{s\sim\rho^{\hat\pi},a\sim\pi}A^{\hat\pi}(s,a) = \frac{\gamma}{1-\gamma}\mathbb{E}_{s,a\sim\rho^{\hat\pi}}[\frac{\pi(a|s)}{\hat\pi(a|s)}-1]\mathbb{E}_{s',a'\sim\rho^\pi(\cdot|s,a)}A^{\hat\pi}(s',a')$$

The proof of the lemma is given in Appendix.

Note from this lemma that the difference between the original formula and the replaced one is relevant to the normalized discount subsequent state-action visitation distribution $\rho^\pi(\cdot|s,a)$. By constraining the right-hand side of the equation, it is easy to obtain Theorem 1 of the paper (Schulman et al., 2015) and Theorem 1 of the paper (Achiam et al., 2017). From this lemma, we constructed a relationship between the current visitation distributions $(s,a)\sim\rho^\pi(\cdot)$ and the next $(s',a')\sim\rho^\pi(\cdot)$.

**Theorem 3.1.** *Consider a current policy $\hat\pi$, and any policies $\pi$, we have*

$$\eta(\pi) = \eta(\hat\pi) + \sum_{i=0}^{k-1}\alpha_i L_i(\pi,\hat\pi) + \beta_k G_k(\pi,\hat\pi) \tag{2}$$

*where*

$$L_i(\pi,\hat\pi) = \mathop{\mathbb{E}}_{\substack{s_0,a_0\sim\rho^{\hat\pi}(\cdot)\\ \cdots \\ s_{i-1},a_{i-1}\sim\rho^{\hat\pi}(\cdot|s_{i-2},a_{i-2})}} \prod_{t=0}^{i-1}(I_t-1)\mathbb{E}_{s_i\sim\rho^{\hat\pi}(\cdot|s_{i-1},a_{i-1}),a_i\sim\pi(\cdot|s_i)}A^{\hat\pi}(s_i,a_i),$$

$$G_k(\pi,\hat\pi) = \mathop{\mathbb{E}}_{\substack{s_0,a_0\sim\rho^{\hat\pi}(\cdot)\\ \cdots \\ s_{k-1},a_{k-1}\sim\rho^{\hat\pi}(\cdot|s_{k-2},a_{k-2})}} \prod_{t=0}^{k-1}(I_t-1)\mathbb{E}_{s_k,a_k\sim\rho^\pi(\cdot|s_{k-1},a_{k-1})}A^{\hat\pi}(s_k,a_k),$$

*and*

$$I_t = \frac{\pi(a_t|s_t)}{\hat\pi(a_t|s_t)}, \ \alpha_i = \frac{\gamma^{i-1}}{(1-\gamma)^i}, \ \beta_k = \frac{\gamma^k}{(1-\gamma)^{k+1}}.$$

The proof of the theorem is given in Appendix.

This theorem gives a general form for the difference between the policy performance of $\pi$ and $\hat\pi$ by finite sums. With this equation, we accurately represent the general gap between the performance of $\pi$ and $\hat\pi$ from a trajectory-based. This portrays that subsequent state-action pairs can also impact optimizing the current policy. We refer to $\sum_{i=0}^k\alpha_i L_i(\pi,\hat\pi)$ as the generalized surrogate objective function. Consider $L_1(\pi,\hat\pi)$ in Eqn. (2) as an example. We consider this function without focusing on the specific form of the parameters. When the environment is unknown, it can only be optimized by sampling. Considering the extreme case, the function $L_1(\pi,\hat\pi)$ is optimized by using a sample $(s_0,a_0,s_1,a_1)$, i.e., $L_1(\pi,\hat\pi) \approx (I_0-1)I_1 A^{\hat\pi}(s_1,a_1)$. If $A^{\hat\pi}(s_1,a_1) < 0$ and $I_0 - 1 < 0$, we know that $(I_0-1)I_1 A^{\hat\pi}(s_1,a_1) = [(I_0-1)A^{\hat\pi}(s_1,a_1)]I_1 > 0$. The probability of $a_1$ is increased. However, when $A^{\hat\pi}(s_1,a_1) < 0$, we should decrease the probability of $a_1$. It's a contradiction. Thus, this term "1" of $I_0 - 1$ may adversely affect policy optimization, though the theory is sound. This situation exists when the environment is unknown. Next, we measure the gap between the policy performance $\eta(\pi)$ and $\sum_{i=0}^k\alpha_i L_i(\pi,\hat\pi)$.

**Corollary 3.1.** *According to the definition of $G_k$, we have*

$$|\beta_k G_k(\pi,\hat\pi)| \le \frac{\gamma^k}{(1-\gamma)^{k+2}}\epsilon^{k+1}R_{\max}$$

*where $\epsilon \triangleq \|\pi-\hat\pi\|_1 = \max_s\sum_a|\pi(a|s)-\hat\pi(a|s)|$ and $R_{\max} \triangleq \max_{s,a}|R(s,a)|$.*

The proof of the theorem is given in the Appendix.

Note that from Theorem 3.1 and Corollary 3.1, the policy performance of $\pi$ has a general lower bound. Compared with Theorem 2 of the paper (Tang et al., 2020), we give a tighter monotonic improvement lower bound (see Appendix). This makes good theoretical sense, which helps the researchers understand the generalized surrogate function. For $k = 1$, the $l_1$ norm constraints are replaced by KL constraints. Further, this result is consistent with the lower bound of TRPO.

## 4 REFLECTIVE POLICY OPTIMIZATION

Theoretically, the previous section gave a tighter lower bound for the policy performance of $\pi$. Although the generalized surrogate function includes the current and subsequent state-action pairs of the trajectory, it is unclear how the subsequent pairs affect the behavior of the policy at the current state, which may have positive or adverse effects. We have slightly modified the generalized surrogate function $L_i(\pi, \hat{\pi})$ of Eqn. (2), and will get the following theorem.

**Theorem 4.1.** *Consider a current policy $\hat{\pi}$, and any policies $\pi$, we have*

$$\eta(\pi) - \eta(\hat{\pi}) \geq \sum_{i=0}^{k-1} \alpha_i \hat{L}_i(\pi, \hat{\pi}) - \hat{C}_k(\pi, \hat{\pi}) \tag{3}$$

*where*

$$\hat{L}_i(\pi, \hat{\pi}) = \mathop{\mathrm{E}}_{\substack{s_0, a_0 \sim \rho^{\hat{\pi}}(\cdot) \\ \cdots \\ s_{i-1}, a_{i-1} \sim \rho^{\hat{\pi}}(\cdot | s_{i-2}, a_{i-2}) \\ s_i, a_i \sim \rho^{\hat{\pi}}(\cdot | s_{i-1}, a_{i-1})}} \prod_{t=0}^{i} I_t A^{\hat{\pi}}(s_i, a_i),$$

$$\hat{C}_k(\pi, \hat{\pi}) = \frac{\gamma R_{\max} I_{k \geq 2}}{(1-\gamma)^2 (1-2\gamma)} \left(1 - \frac{\gamma^k}{(1-\gamma)^k}\right) \|\pi - \hat{\pi}\|_1 + \frac{\gamma^k R_{\max}}{(1-\gamma)^{k+2}} \|\pi - \hat{\pi}\|_1^2,$$

*and $I_{k \geq 2}$ is the indicator function w.r.t. $k \in N$, $\alpha_i = \frac{\gamma^i}{(1-\gamma)^{i+1}}$.*

The proof of the theorem is given in Appendix.

From the theorem 4.1, the first term of the generalized lower bound is called the new generalized surrogate function and the second term is called the penalty term. We know that TRPO (Schulman et al., 2015) is a special case of the generalized lower bound for $k = 1$. Note that improving the surrogate objective can guarantee the improvement of the expected total discounted reward $\eta$. In other words, by optimizing the generalized lower bound, we can get a monotonically improving sequence of policies $\{\pi_i\}_{i=0}^{\infty}$, satisfy $\eta(\pi_0) \leq \eta(\pi_1) \leq \cdots$. Next, we intuitively analyze the new generalized surrogate function. The difference between the function $L_i(\pi, \hat{\pi})$ and $\hat{L}_i(\pi, \hat{\pi})$ is very small, that is, removing the number 1 from the ratios' product. However, the meanings that are intended to be conveyed in them are quite different. We see that if the environment is unknown, the function $L_i(\pi, \hat{\pi})$ may incorrectly optimize the probability of actions (discussed in the previous section). But the function $\hat{L}_i(\pi, \hat{\pi})$ can directly utilize the information between the current and subsequent state-action pairs of the trajectories to optimize the current policy.

With $k = 2$, we will explain in detail. The function $\hat{L}_1(\pi, \hat{\pi})$ contain the ratio of the pair $(s, a)$ and $(s', a')$. If $A^{\hat{\pi}}(s', a') > 0$, one can see that the action $a'$ is fine, then the probability of it will be increased by optimizing the algorithm. At the same time, the state $s'$ is probably fine, too. In order to get into this state again, we should increase the probability of the action $a$ under state $s$. In contrast, if $A^{\hat{\pi}}(s', a') < 0$, the action $a'$ is bad, then the probability of it will be decreased by optimizing the algorithm. Meanwhile, the state $s'$ is probably bad, too. In order to avoid falling back into this state, we should decrease the probability of the action $a$ under state $s$, that is, the agent are able to reflect on current behavior based on subsequent information. For $\hat{L}_0(\pi, \hat{\pi})$, the action's $a$ probability can be optimizing using the advantage function $A^{\hat{\pi}}(s, a)$. Therefore, optimizing the current action $a$ will be influenced by the current and subsequent advantage functions $A^{\hat{\pi}}$ and take them into account. In this way, the optimized policy is likely to have the ability to be reflective for the agent and we can see that optimizing the generalized surrogate function will not have this

ability. Using the same trajectory, more information is learned by the agent. Hence, we explain the whole optimization procedure intuitively. We verify this intuition experimentally. From Figure 2, we conduct the experiment with the CliffWalking environment. Figure 2 shows that optimizing the new surrogate function reduces the number of falling off the Cliff and also faster after reaching the goal $G$. In the experimental section, we explain this phenomenon in detail. The following theorem shows that the modified generalized surrogate function has another nice property except for the monotonicity.

The theorem 4.1 shows that the generalized lower bound is optimized for any $k$. As $k$ increases, the generalized lower bound is optimized using subsequent samples to be able to learn implicit relationships of the current and subsequent states and actions data. But is it suitable when $k$ takes a large value? The answer is no. Let's look at the $\hat{L}_k(\pi, \hat{\pi})$ function individually. This objective function is composed of the product of the $k$ ratios and an advantage function. If the ratio is too much, it faces the problem of high variance (Munos et al., 2016), which in turn affects the stability of the algorithm. In view of this weakness, a very large value of $k$ cannot be taken in practice. In the experimental section, we discuss the values of $k$ and find that as long as the agent makes use of the relationship between before and after state-action pairs, it will enable the agent to fall into the Cliff less often and to reach the goal $G$ faster. The experimental results are similar using either $k = 2$ or $k = 3$. Therefore, the main part of the following is discussed in terms of $k = 2$ , we have $\hat{L}_0(\mu, \hat{\pi}) = \mathbb{E}_{s_0, a_0 \sim \rho^{\hat{\pi}}(\cdot)} I_0 A^{\hat{\pi}}(s_0, a_0)$ and $\hat{L}_1(\mu, \hat{\pi}) = \mathbb{E}_{s_0, a_0 \sim \rho^{\hat{\pi}}(\cdot), s_1, a_1 \sim \rho^{\hat{\pi}}(\cdot | s_0, a_0)} I_0 I_1 A^{\hat{\pi}}(s_1, a_1)$.

**Theorem 4.2.** *For $k = 2$, defined two sets*

$$\Psi_1 = \left\{ \mu \mid \alpha_0 \hat{L}_0(\mu, \hat{\pi}) - \hat{C}_1(\mu, \hat{\pi}) > 0 \right\},$$

$$\Psi_2 = \left\{ \mu \mid \alpha_0 \hat{L}_0(\mu, \hat{\pi}) + \alpha_1 \hat{L}_1(\mu, \hat{\pi}) - \hat{C}_2(\mu, \hat{\pi}) > 0 \right\},$$

*then we have*

$$\Psi_2 \subseteq \Psi_1.$$

The proof of the theorem is given in Appendix.

Note that when $k = 1$, the set $\Psi_1$ is a solution space of TRPO. And when $k > 1$, the set $\Psi_k$ is a solution space of the $k$-th generalized lower bound. The theorem 4.2 shows that the solution space is contracting when $k = 2$. It is also important to note that $\pi^\star$ is in both sets. As shown in Figure 1, reducing the solution space is possibly more efficient in finding a good policy and therefore it is intuitive that the convergence procedure of the algorithm can be accelerated. Similarly, we can define the solution space of $k = 3, 4, \cdots$ and we can use the same way to get $\Psi_1 \supseteq \Psi_2 \supseteq \Psi_3 \supseteq \Psi_4 \supseteq \cdots$. Note that $\pi^\star$ is in those sets. This reveals the benefits of using current and subsequent states and actions of trajectory data to optimize the policy. This provide a promising theoretical basis for our algorithm.

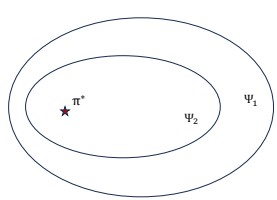

Figure 1: The Solution Space

### 4.1 THE CLIPPED GENERALIZED SURROGATE OBJECTION

In the previous subsection discussion, the generalized lower bound function contained the generalized surrogate function and a penalty term. This lower bound will be optimized in the same way as TRPO, using a linear approximation of the surrogate objective and a quadratic approximation of the penalty term. However it needs to compute the inverse matrix of a quadratic approximation of the penalty term. In particular, the generalized lower bound function also includes the relationship between before and after state-action pairs. It is therefore impractical to solve this. Inspired by PPO (Schulman et al., 2017), a practical variant of TRPO, we propose a new clipped surrogate objection according to Eqn. (3).

When $k = 1$, for $\hat{L}_0(\pi, \hat{\pi})$, we use the PPO's objective function:

$$\hat{L}_0^{clip}(\pi, \hat{\pi}) = \mathbb{E}_{(s,a)} \min \left( I(a|s) A^{\hat{\pi}}(s, a) , \text{clip} \left( I(a|s), 1 - \epsilon, 1 + \epsilon \right) A^{\hat{\pi}}(s, a) \right), \qquad (4)$$

where $I(a|s) = \frac{\pi(a|s)}{\hat{\pi}(a|s)}$, $\epsilon$ is the hyperparameter and we ignore the distribution of $(s, a)$.

---
**Algorithm 1** Reflective Policy Optimization (RPO)
---

Environment $E$, discount factor $\gamma$, batch size $n$, clipping parameter $\epsilon$ and $\epsilon_1$, learning rate $\alpha$.
Initialize policy network parameter $\theta$.
**for** $t = 0, 1, 2, \ldots$ **do**
  Collect data:
  Collect $n$ samples with $\pi_t$ on environment $E$.
  Estimate policy objective:
  Samples a policy data $\pi_t$, estimate on-policy advantage $A^{\pi_t}$ using GAE method, approximately
  estimate maximize the empirical objective $\hat{L}_0^{clip}(\pi, \pi_t)$ and $\hat{L}_1^{clip}(\pi, \pi_t)$ according to Eqn.(4)
  and Eqn.(5).
  The full objective: $\hat{L}(\pi_\theta) \leftarrow \hat{L}_0^{clip}(\pi_\theta, \pi_t) + \beta \hat{L}_1^{clip}(\pi_\theta, \pi_t)$.
  Update policy network:
  Update gradient: $\theta \leftarrow \theta + \alpha \nabla_\theta \hat{L}(\pi_\theta)$.
**end for**

---

When $k = 2$, for $\hat{L}_1(\pi, \hat{\pi})$, we simply modify the clipping mechanism:

$$
\begin{aligned}
\hat{L}_1^{clip}(\pi, \hat{\pi}) = &\mathbb{E}_{(s,a,s',a')} \min \left( I(a|s)I(a'|s')A^{\hat{\pi}}(s', a') \right., \\
& \left. \text{clip}\left(I(a|s), 1 - \epsilon, 1 + \epsilon\right) \text{clip}\left(I(a'|s'), 1 - \epsilon_1, 1 + \epsilon_1\right) A^{\hat{\pi}}(s', a') \right),
\end{aligned}
\tag{5}
$$

where $I(a|s) = \frac{\pi(a|s)}{\hat{\pi}(a|s)}$, $I(a'|s') = \frac{\pi(a'|s')}{\hat{\pi}(a'|s')}$, $\epsilon$ and $\epsilon$ are the hyperparameter and we ignore the distribution of random variables $(s, a, s', a')$.

From the Eqn. (5), we are doing the clipping mechanism for each of the ratios, not all together. If the ratio $I(a|s)$ is large and the ratio $I(a'|s')$ is small, the product of $I(a|s)$ and $I(a'|s')$ may be between $1 - \epsilon$ and $1 + \epsilon$. If their product is clipped, it will continue to optimize the policy and then the result may become better or worse. We have no control over this. Therefore, the way we use the separate clipping mechanism will take into account this unreasonable situation. Through the clipping mechanism, this constrains the variance of the ratio and makes the training procedure of the algorithm more stable. In practice, we find that the parameter $\epsilon_1$ cannot be too big, and it's better to be a little smaller than $\epsilon$. This is because although we want to use the subsequent state-action information to subsidiarily optimize the current policy, and equally don't want the old and new policy to change too quickly. This can once again make the training procedure more stable. When $k > 2$, using the same clipping mechanism approach we can clip the function $\hat{L}_k(\pi, \hat{\pi})$. So, for the generalized lower bound function, we provide a more practice version of the algorithm.

Combining Eqn. (4) and Eqn. (5), we present the Reflective Policy Optimization algorithm (RPO), a practical variant for the generalized surrogate objective function:

$$
\hat{L}(\pi, \pi_t) = \hat{L}_0^{clip}(\pi, \pi_t) + \beta \hat{L}_1^{clip}(\pi, \pi_t)
\tag{6}
$$

where $\hat{L}_0^{clip}(\pi, \pi_t)$ is defined in Eqn.(4), $\hat{L}_1^{clip}(\pi, \pi_t)$ is defined in Eqn.(5), and $\beta > 0$. By choosing the parameter $\beta$, this parameter plays a role in weighting the use of subsequent state-action pair information. Eqn. (6) is the optimization objective function for the $t$-th update. Algorithm 1 shows the detailed implementation pipeline. In each iteration, the RPO algorithm is divided into three steps: collect samples, estimate policy objectives, and update the policy network. It can be seen that our proposed method is also an on-policy algorithm.

**Discuss with multi-step RL**  Multi-step reinforcement learning (RL) is a set of methods that aim to adjust the trade-off of utilization between the knowledge of the current and future return. Recent advances in multi-step RL have achieved remarkable empirical success (Wu et al., 2023; Yunhao et al., 2022). This approach does not directly optimize the current policy but is based on the value function estimated in multi-steps. It is difficult to see directly what role multi-step RL plays in the policy optimization procedure. However, the approach proposed in this paper is viewed from a multi-step perspective: this is directly acting on policy optimization. This will have a direct effect on the actions of the agent. From Figure 2, a clear change in the behavior of the agent can be observed through the visualization, i.e. there is less dropping into the Cliff and reaching goal G more quickly. Therefore, our proposed method is fundamentally different from multi-step RL.

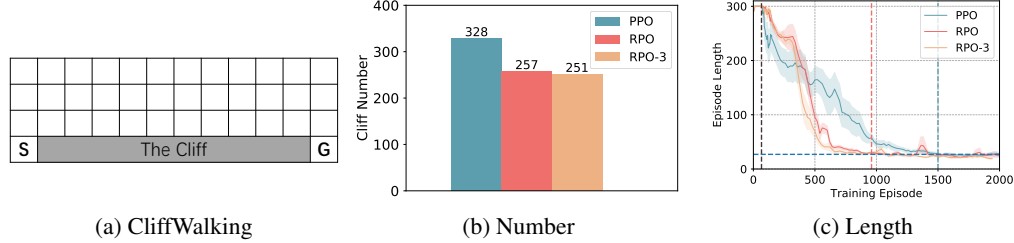

(a) CliffWalking      (b) Number      (c) Length

Figure 2: (a) is a CliffWalking environment. (b) represents the total number of times the agent fell into the Cliff during the training procedure. (c) represents the number of steps taken by the agent to reach the goal $G$ during the training procedure. RPO-3 means that when $k = 3$, the algorithm uses three ratios.

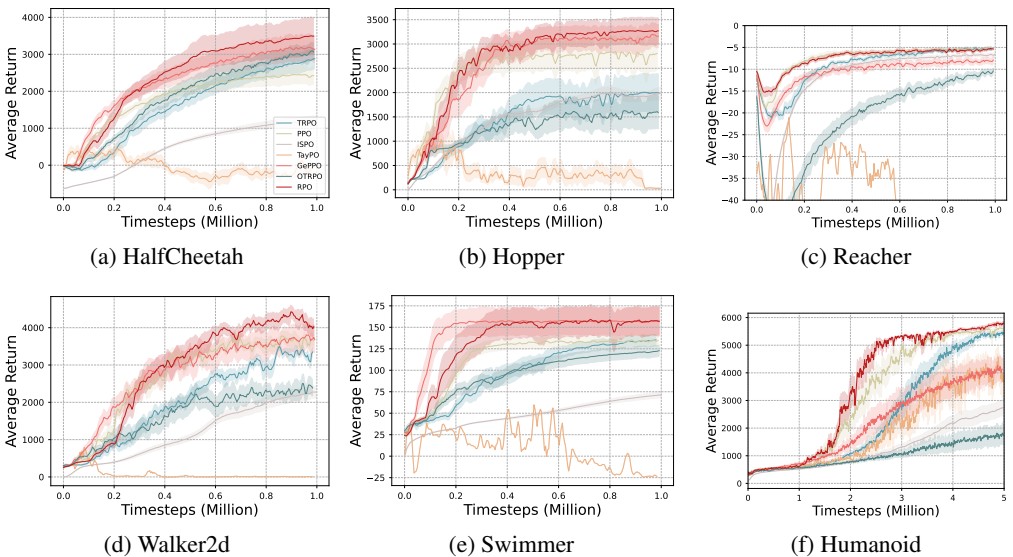

(a) HalfCheetah      (b) Hopper      (c) Reacher

(d) Walker2d      (e) Swimmer      (f) Humanoid

Figure 3: Learning curves on the Gym environments. Performance of RPO vs. PPO, TRPO, OTRPO, GePPO, ISPO and TayPO.

## 5 EXPERIMENTS

### 5.1 VISUAL VALIDATION EXPERIMENT

To demonstrate the effectiveness of the "Reflective Mechanism" of RPO, we conducted visual validation experiments in the CliffWalking environment. CliffWalking is a classic setting widely used for visualizing the performance of reinforcement learning algorithms. Figure 2 illustrates the overall performance of RPO and its baseline algorithm in this test set, especially focusing on the frequency of falling off the cliff and the interaction step overhead, assisting in validating the advantages of RPO's "Reflective Mechanism".

As shown in Figure 2 (b), RPO significantly reduces the frequency of falling off the cliff under equal iteration conditions. This data attests to the significant efficiency of RPO's "Reflective Mechanism". It capitalizes on previous interaction experiences, substantially reducing the occurrence rate of poor decisions. Figure 2 (c) reveals that as the number of interactions increases, RPO markedly cuts down the interaction step overhead per episode, which further confirms the benefits of utilizing the subsequent data. It can also be seen that the number of times RPO and F fall into the cliff and the length of the trajectory are roughly the same. The successful implementation of this mechanism of RPO is attributed to its unique approach to continuous state comprehensive analysis. It's noteworthy that RPO distinguishes itself from the majority of existing algorithms by integrating the strengths

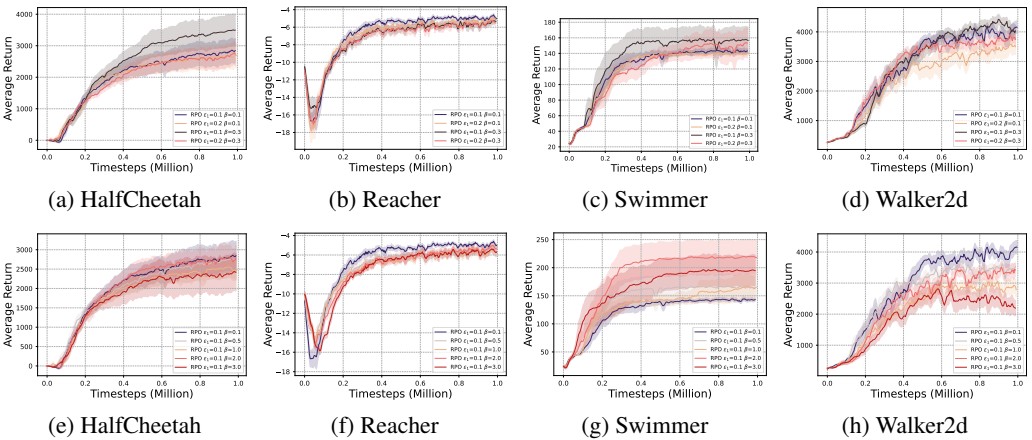

Figure 4: The top line represents the performance under the condition of $\beta$ fixed, and the bottom line represents the performance under the condition of $\epsilon_1$ fixed.

of both current and subsequent data. Unlike other exploitation strategies, RPO efficiently utilizes "good" experiences, makes adjustments based on "bad" experiences, and possesses the ability to predict dynamic changes in the environment. It can more accurately incorporate the development of future states, which is a comprehensive feature that current peer algorithms do not have.

## 5.2 MAIN EXPERIMENT ANALYSIS

To thoroughly validate the extensive effectiveness and universal adaptability of RPO in reinforcement learning scenarios, we conducted six groups of experiments in the continuous action space environment. Since the CliffWalking environment in gym (Brockman et al., 2016) is especially conducive to showcasing RPO's "Reflective Mechanism," we performed auxiliary experiments in this setting.

To thoroughly evaluate the performance of the RPO algorithm, we conducted a detailed comparative analysis with six mainstream algorithms in the field (TRPO (Schulman et al., 2015), PPO (Schulman et al., 2017), GePPO (Queeney et al., 2021), OTRPO (Meng et al., 2022), TayPo (Tang et al., 2020) and ISPO (Tomczak et al., 2019)) in six major experimental environments of MuJoCo (Todorov et al., 2012). The results (as shown in Figure 3) indicate that RPO consistently outperforms in all MuJoCo sub-environments.

In these six diverse testing environments, RPO surpasses classic on-policy reinforcement learning algorithms PPO and TRPO not only in terms of average return but also in convergence speed. This improvement is attributed to RPO's incorporation of the strengths of both current and subsequent data. When compared to the enhanced off-policy algorithms OTRPO and GePPO, RPO also exhibits significant advantages.

The exceptional performance of RPO is rooted in its unique Reflective mechanism that facilitates the efficient utilization of both positive and negative experiences. By employing short trajectories composed of two consecutive states for learning and decision-making, a more profound reflection and utilization of experience is achieved. This approach has the following benefits: it enables the effective use of interaction experiences from adjacent states. And by adopting this pair-wise state combination for short trajectory inputs, computational and storage overhead is minimized while maximizing the retention of temporality, promoting a deeper utilization of experience and Reflective mechanism.

From the analysis above, it is evident that RPO exhibits significant advantages in various aspects, especially in convergence speed and average return, compared to other algorithms. These empirical findings underscore the efficiency and applicability of the RPO algorithm in complex continuous action space environments.

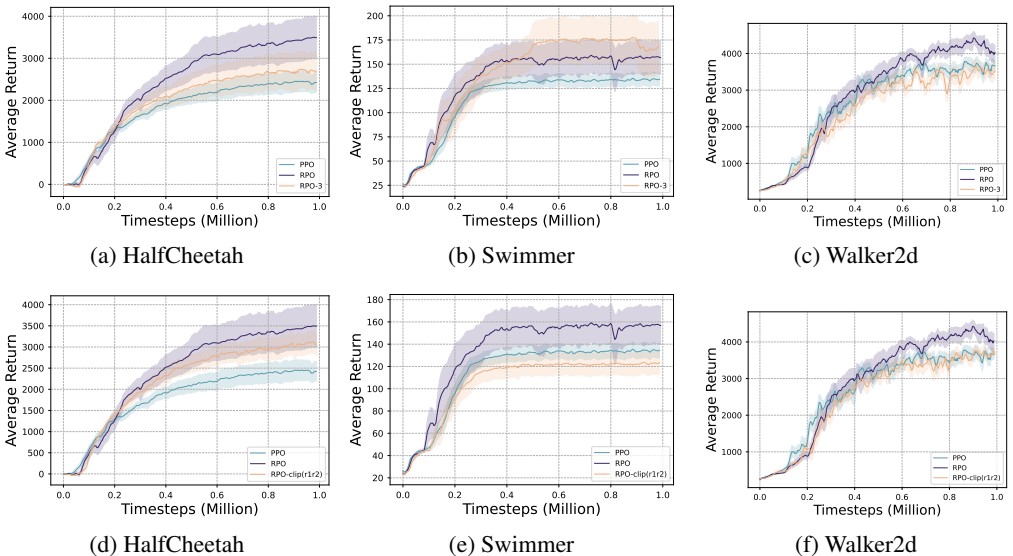

(a) HalfCheetah      (b) Swimmer      (c) Walker2d

(d) HalfCheetah      (e) Swimmer      (f) Walker2d

Figure 5: The top line represents the performance of RPO vs. RPO-3 (this means that when $k = 3$, the algorithm uses three ratios), and the bottom line represents the performance of RPO vs. RPO-clip(r1r2)( this means that the two ratios are clipped together.).

### 5.3 ABLATION EXPERIMENT ANALYSIS

Initially, we focused on the clip and weighting coefficients, applying parameter ablation in the RPO. The objective of these two clips is to maintain the stability of the policy, addressing the risk associated with the current methods that only clip the product, which can lead to an imbalance in the proportion of the two factors and subsequently, an unstable policy update. As can be inferred from Figure 4 (a-d), under a fixed weighting coefficient, the smaller the clip, the more pronounced the results. Furthermore, Figure 4 (e-h) reveals that with a certain clip, reducing the weighting coefficient within an acceptable range positively influences the outcome. In essence, the impact varies with different clips and weighting coefficients, yet all outcomes under every parameter exceed the baseline. These results reconfirm the indispensability of the RPO algorithm's introspection mechanism. It fosters in-depth experiential learning from new short trajectories formed by preceding and succeeding states, promoting stable and enhanced performance, and expediting the model's convergence rate.

Secondly, we conducted an ablation study on the number of states, as shown in Figure 5. The results reveal that selecting three states with sequential relationships yields an equivalent effect to choosing two states. Thus, we think that the number of trajectories in RPO need not exceed two, as two states suffice for effective reflection.

Finally, we conducted ablation experiments on whether the two ratios were clipped together or not, and by comparing it with the RPO clip(r1r2) (this means that the two ratios are clipped together) algorithm, it is evident that my formula exhibits greater performance, as indicated by the results.

## 6 CONCLUSION

In this paper, we propose a simple on-policy algorithm, called Reflective Policy Optimization (RPO). This method aims to combine the previous and next state and action information of the trajectory data to optimize the current policy, thus allowing the agent to reflect on and modify the action of the current state to some extent. Furthermore, theoretical analyses show that our proposed method, in addition to satisfying the desirable property of the monotonic improvement of policy performance, can effectively reduce the solution space of the optimized policy, resulting in speeding up the training procedure of the algorithm. we verify the feasibility and effectiveness of the proposed method by a toy example and achieve better performance on RL benchmarks

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

## A APPENDIX

### A.1 PROOF

Let's start with some useful lemmas.

**Lemma A.1.** *((Kakade & Langford, 2002)) Consider any two policies $\hat{\pi}$ and $\pi$, we have*

$$\eta(\pi) - \eta(\hat{\pi}) = \frac{1}{1-\gamma} \mathbb{E}_{s,a \sim \rho^\pi} A^{\hat{\pi}}(s,a).$$

**Corollary A.1.** *Consider any two policies $\hat{\pi}$ and $\pi$, we have*

- $V^\pi(s_0) - V^{\hat{\pi}}(s_0) = \frac{1}{1-\gamma} \mathbb{E}_{s,a \sim \rho^\pi(\cdot|s_0)} A^{\hat{\pi}}(s,a).$

- $Q^\pi(s_0, a_0) - Q^{\hat{\pi}}(s_0, a_0) = \frac{\gamma}{1-\gamma} \mathbb{E}_{s,a \sim \rho^\pi(\cdot|s_0,a_0)} A^{\hat{\pi}}(s,a).$

*Proof.* The first formula is simple, due to $\eta(\pi) = \mathbb{E}_{s_0 \sim \rho_0} V^\pi(s_0)$.

Let's prove the second formula.

$$\begin{aligned}
&Q^\pi(s_0, a_0) - Q^{\hat{\pi}}(s_0, a_0) \\
=&\gamma \mathbb{E}_{s' \sim P(s'|s_0,a_0)} \left[ V^\pi(s') - V^{\hat{\pi}}(s') \right] \\
=&\frac{\gamma}{1-\gamma} \mathbb{E}_{s' \sim P(s'|s_0,a_0)} \mathbb{E}_{s,a \sim \rho^\pi(\cdot|s')} A^{\hat{\pi}}(s,a) \\
=&\frac{\gamma}{1-\gamma} \mathbb{E}_{s,a \sim \rho^\pi(\cdot|s_0,a_0)} A^{\hat{\pi}}(s,a).
\end{aligned}$$

$\square$

**Lemma A.2.** *((Tomczak et al., 2019)) Consider any two policies $\hat{\pi}$ and $\pi$, we have*

$$\eta(\pi) - \eta(\hat{\pi}) = \frac{1}{1-\gamma} \mathbb{E}_{s \sim \rho^{\hat{\pi}}, a \sim \pi} A^{\hat{\pi}}(s,a) + \frac{1}{1-\gamma} \mathbb{E}_{s,a \sim \rho^{\hat{\pi}}} \left[ \frac{\pi(a|s)}{\hat{\pi}(a|s)} - 1 \right] \left[ Q^\pi(s,a) - Q^{\hat{\pi}}(s,a) \right]$$

**Lemma A.3.** *Consider a current policy $\hat{\pi}$, and any policies $\pi$, we have*

$$\begin{aligned}
&\mathbb{E}_{s,a \sim \rho^\pi(\cdot)} A^{\hat{\pi}}(s,a) - \mathbb{E}_{s \sim \rho^{\hat{\pi}}, a \sim \pi} A^{\hat{\pi}}(s,a) \\
=&\frac{\gamma}{1-\gamma} \mathop{\mathbb{E}}_{\substack{s,a \sim \rho^{\hat{\pi}}(\cdot) \\ s',a' \sim \rho^\pi(\cdot|s,a)}} [\frac{\pi(a|s)}{\hat{\pi}(a|s)} - 1] A^{\hat{\pi}}(s',a')
\end{aligned}$$

*Proof.* From Lemma A.1 and A.2, we have

$$\begin{aligned}
&\mathbb{E}_{s,a \sim \rho^\pi} A^{\hat{\pi}}(s,a) - \mathbb{E}_{s \sim \rho^{\hat{\pi}}, a \sim \pi} A^{\hat{\pi}}(s,a) \\
=&\mathbb{E}_{s,a \sim \rho^{\hat{\pi}}} \left[ \frac{\pi(a|s)}{\hat{\pi}(a|s)} - 1 \right] \left[ Q^\pi(s,a) - Q^{\hat{\pi}}(s,a) \right]
\end{aligned}$$

According to Corollary A.1, it is easy to get the conclusion. $\square$

**Theorem A.1.** *Consider a current policy $\hat{\pi}$, and any policies $\pi$, we have*

$$\eta(\pi) = \eta(\hat{\pi}) + \sum_{i=0}^{k-1} \alpha_i L_i(\pi, \hat{\pi}) + \beta_k G_k(\pi, \hat{\pi})$$

*where*

$$L_i(\pi, \hat{\pi}) = \underset{\substack{s_0, a_0 \sim \rho^{\hat{\pi}}(\cdot) \\ \cdots \\ s_{i-1}, a_{i-1} \sim \rho^{\hat{\pi}}(\cdot | s_{i-2}, a_{i-2})}}{\mathrm{E}} \prod_{t=0}^{i-1} (I_t - 1) l_i(\pi, \hat{\pi}),$$

$$G_k(\pi, \hat{\pi}) = \underset{\substack{s_0, a_0 \sim \rho^{\hat{\pi}}(\cdot) \\ \cdots \\ s_{k-1}, a_{k-1} \sim \rho^{\hat{\pi}}(\cdot | s_{k-2}, a_{k-2})}}{\mathrm{E}} \prod_{t=0}^{k-1} (I_t - 1) g_k(\pi, \hat{\pi}),$$

$$l_i(\pi, \hat{\pi}) = \mathbb{E}_{s_i \sim \rho^{\hat{\pi}}(\cdot | s_{i-1}, a_{i-1}), a_i \sim \pi(\cdot | s_i)} A^{\hat{\pi}}(s_i, a_i),$$

$$g_k(\pi, \hat{\pi}) = \mathbb{E}_{s_k, a_k \sim \rho^{\pi}(\cdot | s_{k-1}, a_{k-1})} A^{\hat{\pi}}(s_k, a_k),$$

*and*

$$I_t = \frac{\pi(a_t | s_t)}{\hat{\pi}(a_t | s_t)}, \ \alpha_i = \frac{\gamma^i}{(1-\gamma)^{i+1}}, \ \beta_k = \frac{\gamma^k}{(1-\gamma)^{k+1}}.$$

*Proof.* From Lemma A.3, this formula creates a link between $\mathbb{E}_{s,a \sim \rho^{\pi}(\cdot)} A^{\hat{\pi}}(s,a)$ and $\mathbb{E}_{s', a' \sim \rho^{\pi}(\cdot | s, a)} A^{\hat{\pi}}(s', a')$, resulting in a recursive relationship.

According to Lemma A.2, and using recursive relationships, defined

$$l_i(\pi, \hat{\pi}) = \mathbb{E}_{s_i \sim \rho^{\hat{\pi}}(\cdot | s_{i-1}, a_{i-1}), a_i \sim \pi(\cdot | s_i)} A^{\hat{\pi}}(s_i, a_i),$$

we have

$$\eta(\pi) - \eta(\hat{\pi})$$

$$= \frac{1}{1-\gamma} \mathbb{E}_{s_0 \sim \rho^{\hat{\pi}}, a_0 \sim \pi} A^{\hat{\pi}}(s_0, a_0) + \frac{\gamma}{(1-\gamma)^2} \mathbb{E}_{s_0, a_0 \sim \rho^{\hat{\pi}}} [I_0 - 1] \mathbb{E}_{s_1, a_1 \sim \rho^{\pi}(\cdot | s_0, a_0)} A^{\hat{\pi}}(s_1, a_1)$$

$$= \frac{1}{1-\gamma} l_0(\pi, \hat{\pi})$$

$$+ \frac{\gamma}{(1-\gamma)^2} \mathbb{E}_{s_0, a_0 \sim \rho^{\hat{\pi}}} [I_0 - 1] \left( l_1(\pi, \hat{\pi}) + \frac{\gamma}{1-\gamma} \mathbb{E}_{s_1, a_1 \sim \rho^{\hat{\pi}}(\cdot | s_0, a_0)} [I_1 - 1] \mathbb{E}_{s_2, a_2 \sim \rho^{\pi}(\cdot | s_1, a_1)} A^{\hat{\pi}}(s_2, a_2) \right)$$

$$= \frac{1}{1-\gamma} l_0(\pi, \hat{\pi}) + \frac{\gamma}{(1-\gamma)^2} \mathbb{E}_{s_0, a_0 \sim \rho^{\hat{\pi}}} [I_0 - 1] l_1(\pi, \hat{\pi})$$

$$+ \frac{\gamma^2}{(1-\gamma)^3} \underset{\substack{s_0, a_0 \sim \rho^{\hat{\pi}}(\cdot) \\ s_1, a_1 \sim \rho^{\hat{\pi}}(\cdot | s_0, a_0) \\ s_2, a_2 \sim \rho^{\pi}(\cdot | s_1, a_1)}}{\mathrm{E}} [I_0 - 1][I_1 - 1] A^{\hat{\pi}}(s_2, a_2)$$

$$\cdots$$

$$= \sum_{i=0}^{k-1} \alpha_i L_i(\pi, \hat{\pi}) + \beta_k G_k(\pi, \hat{\pi})$$

$\square$

**Corollary A.2.** *According to the definition of $G_k$, we have*

$$|\beta_k G_k(\pi, \hat{\pi})| \le \frac{\gamma^k}{(1-\gamma)^{k+2}} \epsilon^{k+1} R_{\max},$$

*where $\epsilon \triangleq \|\pi - \hat{\pi}\|_1 = \max_s \sum_a |\pi(a|s) - \hat{\pi}(a|s)|$ and $R_{\max} \triangleq \max_{s,a} |R(s,a)|$.*

*Proof.* According to the definition of $G_k(\pi, \hat{\pi})$, and defined $\epsilon \triangleq \|\pi - \hat{\pi}\|_1$, we have

$$|G_k(\pi, \hat{\pi})| \le \epsilon^k \cdot |\mathbb{E}_{s_k, a_k \sim \rho^{\pi}(\cdot | s_{k-1}, a_{k-1})} A^{\hat{\pi}}(s_k, a_k)|$$

$$\le \epsilon^k \cdot |\int_a (\pi - \hat{\pi}) Q^{\hat{\pi}}(s,a) da|$$

$$\le \frac{R_{\max}}{1-\gamma} \epsilon^{k+1}$$

Combining with $\beta_k$, we can get this conclusion. $\qquad\square$

**Corollary A.3.** *Compared with Theorem 2 of the paper (Tang et al., 2020), we give a tighter lower bound.*

*Proof.* From the paper (Tang et al., 2020), they give the gap between the policy performance of $\pi$ and the general surrogate object

$$\hat{G}_k = \frac{1}{\gamma(1-\gamma)}\left(1 - \frac{\gamma}{1-\gamma}\epsilon\right)^{-1}\left(\frac{\gamma\epsilon}{1-\gamma}\right)^{K+1}R_{\max}$$

Next, from Corollary A.2, we will prove that the following inequality holds

$$\frac{\gamma^k}{(1-\gamma)^{k+2}}\epsilon^{k+1}R_{\max} < \hat{G}_k.$$

That is, we need to prove

$$\frac{\gamma^k}{(1-\gamma)^{k+2}}\epsilon^{k+1}R_{\max} < \frac{1}{\gamma(1-\gamma)}\left(1 - \frac{\gamma}{1-\gamma}\epsilon\right)^{-1}\left(\frac{\gamma\epsilon}{1-\gamma}\right)^{K+1}R_{\max}$$

After simplification, we get

$$\frac{1}{1-\gamma} < \frac{1}{1-\gamma-\gamma\epsilon}.$$

The inequality obviously holds. So, we give a tighter lower bound. $\qquad\square$

**Theorem A.2.** *Consider a current policy $\hat{\pi}$, and any policies $\pi$, we have*

$$\eta(\pi) - \eta(\hat{\pi}) \geq \sum_{i=0}^{k-1}\alpha_i\hat{L}_i(\pi,\hat{\pi}) - \hat{C}_k(\pi,\hat{\pi})$$

*where*

$$\hat{L}_i(\pi,\hat{\pi}) = \mathop{\mathrm{E}}_{\substack{s_0,a_0\sim\rho^{\hat{\pi}}(\cdot)\\ \cdots\\ s_{i-1},a_{i-1}\sim\rho^{\hat{\pi}}(\cdot|s_{i-2},a_{i-2})\\ s_i,a_i\sim\rho^{\hat{\pi}}(\cdot|s_{i-1},a_{i-1})}}\prod_{t=0}^{i}I_t A^{\hat{\pi}}(s_i,a_i),$$

$$\hat{C}_k(\pi,\hat{\pi}) = \frac{\gamma R_{\max}I_{k\geq 2}}{(1-\gamma)^2(1-2\gamma)}\left(1 - \frac{\gamma^k}{(1-\gamma)^k}\right)\|\pi - \hat{\pi}\|_1 + \frac{\gamma^k R_{\max}}{(1-\gamma)^{k+2}}\|\pi - \hat{\pi}\|_1^2$$

*and $I_{k\geq 2}$ is the indicator function w.r.t. $k \in N$ , $\alpha_i = \frac{\gamma^i}{(1-\gamma)^{i+1}}$.*

*Proof.* For the definition of $L_i(\pi, \hat{\pi})$, we have

$$\eta(\pi) - \eta(\hat{\pi})$$

$$=\frac{1}{1-\gamma}\mathbb{E}_{s_0\sim\rho^{\hat{\pi}},a_0\sim\pi}A^{\hat{\pi}}(s_0,a_0) + \frac{\gamma}{(1-\gamma)^2}\mathbb{E}_{s_0,a_0\sim\rho^{\hat{\pi}}}[I_0-1]\mathbb{E}_{s_1,a_1\sim\rho^{\pi}(\cdot|s_0,a_0)}A^{\hat{\pi}}(s_1,a_1)$$

$$=\frac{1}{1-\gamma}l_0(\pi,\hat{\pi}) - \frac{\gamma}{(1-\gamma)^2}\mathbb{E}_{s_0,a_0\sim\rho^{\hat{\pi}}}\mathbb{E}_{s_1,a_1\sim\rho^{\pi}(\cdot|s_0,a_0)}A^{\hat{\pi}}(s_1,a_1)$$

$$+ \frac{\gamma}{(1-\gamma)^2}\mathbb{E}_{s_0,a_0\sim\rho^{\hat{\pi}}}I_0\left(l_1(\pi,\hat{\pi}) + \frac{\gamma}{1-\gamma}\mathbb{E}_{s_1,a_1\sim\rho^{\hat{\pi}}(\cdot|s_0,a_0)}[I_1-1]\mathbb{E}_{s_2,a_2\sim\rho^{\pi}(\cdot|s_1,a_1)}A^{\hat{\pi}}(s_2,a_2)\right)$$

$$=\frac{1}{1-\gamma}l_0(\pi,\hat{\pi}) - \frac{\gamma}{(1-\gamma)^2}\mathbb{E}_{s_0,a_0\sim\rho^{\hat{\pi}}}\mathbb{E}_{s_1,a_1\sim\rho^{\pi}(\cdot|s_0,a_0)}A^{\hat{\pi}}(s_1,a_1)$$

$$+ \frac{\gamma}{(1-\gamma)^2}\mathbb{E}_{s_0,a_0\sim\rho^{\hat{\pi}}}I_0 l_1(\pi,\hat{\pi}) - \frac{\gamma^2}{(1-\gamma)^3}\mathbb{E}_{s_0,a_0\sim\rho^{\hat{\pi}}}I_0\mathbb{E}_{s_1,a_1\sim\rho^{\hat{\pi}}(\cdot|s_0,a_0)}\mathbb{E}_{s_2,a_2\sim\rho^{\pi}(\cdot|s_1,a_1)}A^{\hat{\pi}}(s_2,a_2)$$

$$+ \frac{\gamma^2}{(1-\gamma)^3}\mathbb{E}_{s_0,a_0\sim\rho^{\hat{\pi}}}I_0\mathbb{E}_{s_1,a_1\sim\rho^{\hat{\pi}}(\cdot|s_0,a_0)}I_1\mathbb{E}_{s_2,a_2\sim\rho^{\pi}(\cdot|s_1,a_1)}A^{\hat{\pi}}(s_2,a_2)$$

$$=\frac{1}{1-\gamma}l_0(\pi,\hat{\pi}) + \frac{\gamma}{(1-\gamma)^2}\mathbb{E}_{s_0,a_0\sim\rho^{\hat{\pi}}}[I_0-1]l_1(\pi,\hat{\pi})$$

$$+ \frac{\gamma^2}{(1-\gamma)^3}\mathop{\mathbb{E}}_{\substack{s_0,a_0\sim\rho^{\hat{\pi}}(\cdot)\\s_1,a_1\sim\rho^{\hat{\pi}}(\cdot|s_0,a_0)\\s_2,a_2\sim\rho^{\pi}(\cdot|s_1,a_1)}}[I_0-1][I_1-1]A^{\hat{\pi}}(s_2,a_2)$$

$$\cdots$$

$$=\sum_{i=0}^{k-1}\alpha_i\hat{L}_i(\pi,\hat{\pi}) - \sum_{i=1}^{k-1}\alpha_i\hat{H}_i(\pi,\hat{\pi}) + \beta_k\hat{G}_k(\pi,\hat{\pi})$$

where

$$\hat{L}_i(\pi,\hat{\pi}) = \mathop{\mathbb{E}}_{\substack{s_0,a_0\sim\rho^{\hat{\pi}}(\cdot)\\\cdots\\s_{i-1},a_{i-1}\sim\rho^{\hat{\pi}}(\cdot|s_{i-2},a_{i-2})\\s_i,a_i\sim\rho^{\hat{\pi}}(\cdot|s_{i-1},a_{i-1})}}\prod_{t=0}^{i}I_t A^{\hat{\pi}}(s_i,a_i),$$

$$\hat{H}_i(\pi,\hat{\pi}) = \mathop{\mathbb{E}}_{\substack{s_0,a_0\sim\rho^{\hat{\pi}}(\cdot)\\\cdots\\s_{i-1},a_{i-1}\sim\rho^{\hat{\pi}}(\cdot|s_{i-2},a_{i-2})\\s_i,a_i\sim\rho^{\pi}(\cdot|s_{i-1},a_{i-1})}}\prod_{t=0}^{i-2}I_t A^{\hat{\pi}}(s_i,a_i),$$

$$\hat{G}_k(\pi,\hat{\pi}) = \mathop{\mathbb{E}}_{\substack{s_0,a_0\sim\rho^{\hat{\pi}}(\cdot)\\\cdots\\s_{i-1},a_{i-1}\sim\rho^{\hat{\pi}}(\cdot|s_{i-2},a_{i-2})\\s_i,a_i\sim\rho^{\pi}(\cdot|s_{i-1},a_{i-1})}}\prod_{t=0}^{i-2}I_t[I_{i-1}-1]A^{\hat{\pi}}(s_i,a_i),$$

and $\alpha_i = \frac{\gamma^i}{(1-\gamma)^{i+1}}$, $\beta_k = \frac{\gamma^k}{(1-\gamma)^{k+1}}$.

It is easy to prove that the following inequality holds

$$\hat{H}_i(\pi,\hat{\pi}) \leq \frac{R_{\max}}{1-\gamma}\|\pi-\hat{\pi}\|_1, \ \hat{G}_k(\pi,\hat{\pi}) \leq \frac{R_{\max}}{1-\gamma}\|\pi-\hat{\pi}\|_1^2.$$

Since $\sum_{k-1}^{i=0}\alpha_i = \frac{\gamma}{(1-\gamma)(1-2\gamma)}\left(1 - \frac{\gamma^k}{(1-\gamma)^k}\right)$, we have

$$\eta(\pi) - \eta(\hat{\pi}) \geq \sum_{i=0}^{k-1}\alpha_i\hat{L}_i(\pi,\hat{\pi}) - \frac{\gamma R_{\max}I_{k\geq 2}\|\pi-\hat{\pi}\|_1}{(1-\gamma)^2(1-2\gamma)}\left(1 - \frac{\gamma^k}{(1-\gamma)^k}\right) - \frac{\gamma^k R_{\max}}{(1-\gamma)^{k+2}}\|\pi-\hat{\pi}\|_1^2$$

$$\square$$

**Theorem A.3.** *Define two sets*

$$\Psi_1 = \left\{ \mu \mid \alpha_0 \hat{L}_0(\mu, \hat{\pi}) - \hat{C}_1(\mu, \hat{\pi}) > 0 \right\},$$

$$\Psi_2 = \left\{ \mu \mid \alpha_0 \hat{L}_0(\mu, \hat{\pi}) + \alpha_1 \hat{L}_1(\mu, \hat{\pi}) - \hat{C}_2(\mu, \hat{\pi}) > 0 \right\},$$

*then we have*

$$\Psi_2 \subseteq \Psi_1.$$

*Proof.* Let $\mu \in \Psi_1$, we have

$$\hat{L}_0(\pi, \hat{\pi}) - \frac{\gamma R_{\max}}{(1-\gamma)^2} \|\mu - \hat{\pi}\|_1^2 > 0 \tag{7}$$

Below, we will show that $\mu$ may not be in the set $\Psi_2$.

For $\hat{L}_1(\pi, \hat{\pi})$, we can get

$$\hat{L}_1(\pi, \hat{\pi}) = \mathop{\mathrm{E}}_{\substack{s_0 \sim \rho^{\hat{\pi}}(\cdot), a_0 \sim \pi(\cdot|s_0) \\ s_1 \sim \rho^{\hat{\pi}}(\cdot|s_0, a_0), a_1 \sim \pi(\cdot|s_1)}} A^{\hat{\pi}}(s_1, a_1) \tag{8}$$

$$= \mathop{\mathrm{E}}_{\substack{s_0 \sim \rho^{\hat{\pi}}(\cdot), a_0 \sim \hat{\pi}(\cdot|s_0) \\ s_1 \sim \rho^{\hat{\pi}}(\cdot|s_0, a_0), a_1 \sim \pi(\cdot|s_1)}} A^{\hat{\pi}}(s_1, a_1) + \left( \mathop{\mathrm{E}}_{\substack{s_0 \sim \rho^{\hat{\pi}}(\cdot), a_0 \sim \pi(\cdot|s_0) \\ s_1 \sim \rho^{\hat{\pi}}(\cdot|s_0, a_0), a_1 \sim \pi(\cdot|s_1)}} - \mathop{\mathrm{E}}_{\substack{s_0 \sim \rho^{\hat{\pi}}(\cdot), a_0 \sim \hat{\pi}(\cdot|s_0) \\ s_1 \sim \rho^{\hat{\pi}}(\cdot|s_0, a_0), a_1 \sim \pi(\cdot|s_1)}} \right) A^{\hat{\pi}}(s_1, a_1) \tag{9}$$

$$\geq \mathop{\mathrm{E}}_{s_1 \sim \rho^{\hat{\pi}}(\cdot), a_1 \sim \pi(\cdot|s_1)} A^{\hat{\pi}}(s_1, a_1) - \frac{R_{\max}}{1-\gamma} \|\pi - \hat{\pi}\|_1^2 \tag{10}$$

The last inequality uses $\mathrm{E}_{s_0 \sim \rho^{\hat{\pi}}(\cdot), a_0 \sim \hat{\pi}(\cdot|s_0)} \rho^{\hat{\pi}}(\cdot|s_0, a_0) = \rho^{\hat{\pi}}(\cdot)$ and Hölder's inequality (Finner, 1992).

Combining with $\hat{L}_0(\pi, \hat{\pi})$ and $\hat{C}_2(\pi, \hat{\pi})$, we have

$$\hat{L}_0(\pi, \hat{\pi}) + \frac{\gamma}{1-\gamma} \hat{L}_1(\pi, \hat{\pi}) - \frac{\gamma R_{\max}}{(1-\gamma)^2} \|\pi - \hat{\pi}\|_1 - \frac{\gamma^2 R_{\max}}{(1-\gamma)^3} \|\pi - \hat{\pi}\|_1^2$$

$$\geq \hat{L}_0(\pi, \hat{\pi}) + \frac{\gamma}{1-\gamma} \left( \mathop{\mathrm{E}}_{s_1 \sim \rho^{\hat{\pi}}(\cdot), a_1 \sim \pi(\cdot|s_1)} A^{\hat{\pi}}(s_1, a_1) - \frac{R_{\max}}{1-\gamma} \|\pi - \hat{\pi}\|_1^2 \right) - \frac{\gamma R_{\max}}{(1-\gamma)^2} \|\pi - \hat{\pi}\|_1 - \frac{\gamma^2 R_{\max}}{(1-\gamma)^3} \|\pi - \hat{\pi}\|_1^2$$

$$= \frac{1}{1-\gamma} \hat{L}_0(\pi, \hat{\pi}) - \frac{\gamma R_{\max}}{(1-\gamma)^3} \|\pi - \hat{\pi}\|_1^2 - \frac{\gamma R_{\max}}{(1-\gamma)^2} \|\pi - \hat{\pi}\|_1$$

Combining with the inequality (7), we have

$$\hat{L}_0(\mu, \hat{\pi}) + \frac{\gamma}{1-\gamma} \hat{L}_1(\mu, \hat{\pi}) - \frac{\gamma R_{\max}}{(1-\gamma)^2} \|\mu - \hat{\pi}\|_1 - \frac{\gamma^2 R_{\max}}{(1-\gamma)^3} \|\mu - \hat{\pi}\|_1^2 \geq - \frac{\gamma R_{\max}}{(1-\gamma)^2} \|\mu - \hat{\pi}\|_1$$

From the above inequality, it shows that $\mu$ may not be in set $\Psi_2$. So, we have $\Psi_2 \subseteq \Psi_1$. $\qquad\square$

## A.2 ADDITIONAL EXPERIMENTAL RESULTS

To verify the effectiveness of the proposed RPO method, we select six continuous control tasks from the MuJoCo environments (Todorov, Erez, and Tassa 2012) in OpenAI Gym (Brockman et al. 2016). We conduct all the experiments mainly based on the code from (Queeney, Paschalidis, and Cassandras 2021). The test procedures are averaged over ten test episodes across ten independent runs. The same neural network architecture is used for all methods The policy network is a Gaussian distribution, and the output of the state-value network is a scalar value. The mean action of the policy network and state-value network are a multi-layer perceptron with hidden layer fixed to [64, 64] and tanh activation (Henderson et al. 2018). The standard deviation of the policy network is

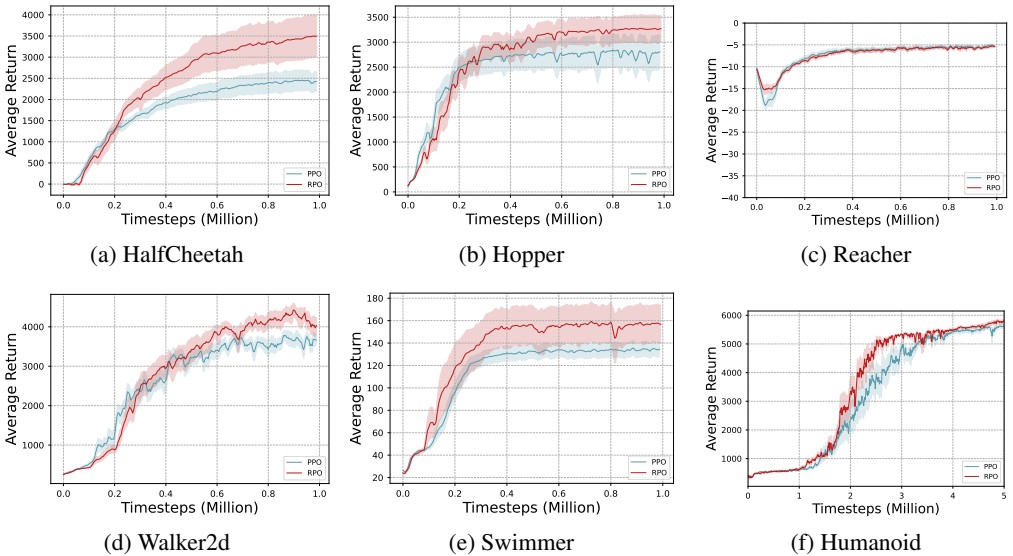

Figure 6: Learning curves on the Gym environments. Performance of RPO vs. PPO.

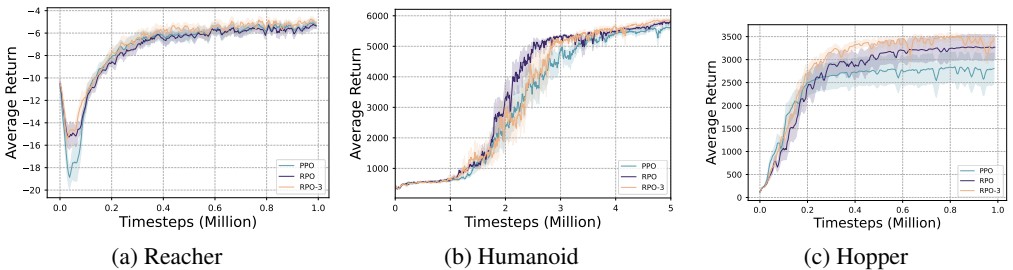

Figure 7: The performance of RPO vs. RPO-3 in three other environments.

parameterized separately (Schulman et al. 2015, 2017). For the experimental parameters, we use the default parameters from (Dhariwal et al. 2017; Henderson et al. 2018), for example, the discount factor is $\gamma = 0.995$, and we use the Adam optimizer (Diederik et al. 2015) throughout the training progress. For PPO, the clipping parameter is $\epsilon^{\text{PPO}} = 0.2$, and the batch size is $B = 2048$. For GePPO, the clipping parameter is $\epsilon^{\text{PPO}} = 0.1$, and the batch size of each policy is $B = 1024$. For TRPO and off-policy TRPO (OTRPO), the bound of trust region is $\delta = 0.01$, and the batch size of each policy is $B = 1024$.

To verify the effectiveness of the proposed RPO method in discrete environments, we randomly selected twelve Atari games for our experiments and the code is based on (Zhang, 2018). We run our experiments across three seeds with fair evaluation metrics. We use the same hyperparameters $\epsilon_1 = 0.1$ and $\beta = 3.0$ in all environments and do not fine-tune them.

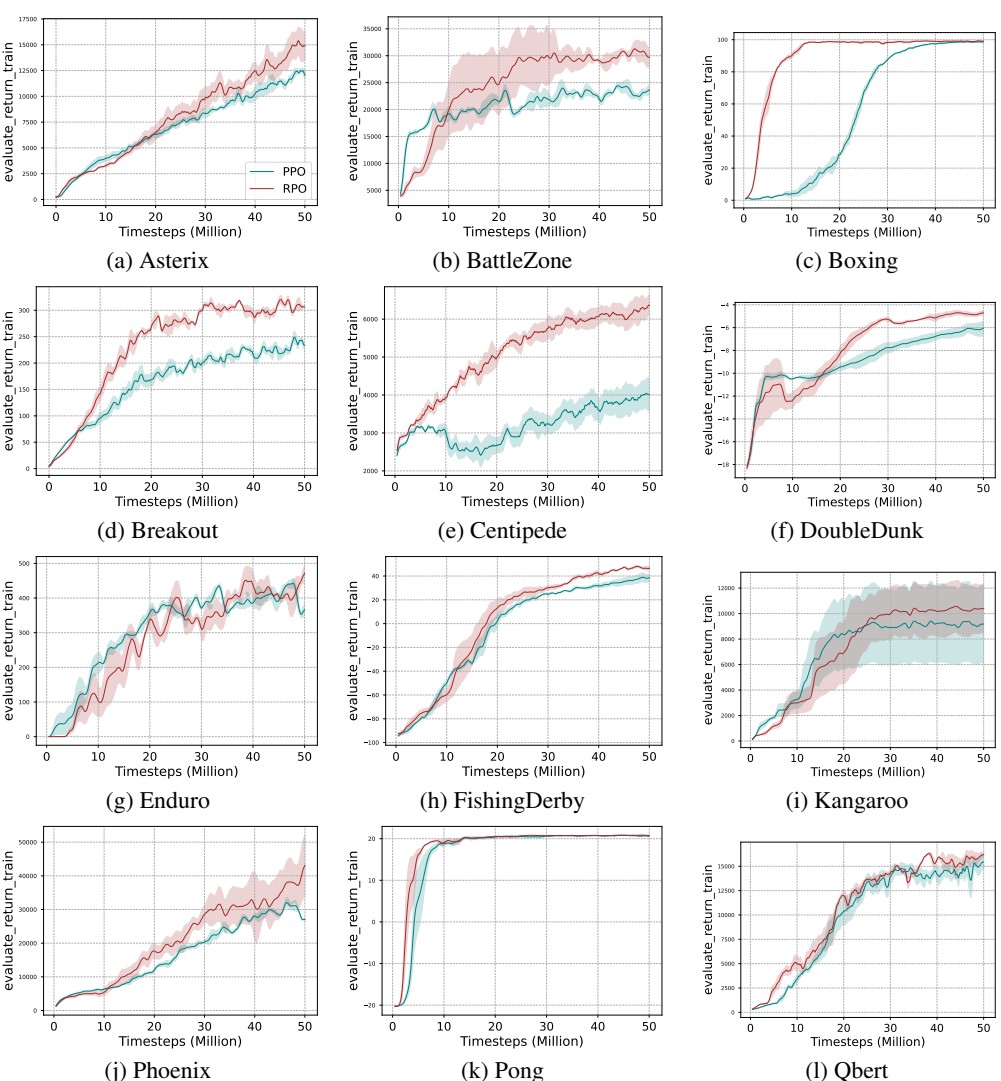

Figure 8: Learning curves on the Atari environments. Performance of RPO vs. PPO.

Table 1: Hyperparameters for RPO on Mujoco tasks.

| Hyperparameter | Value |
|---|---|
| Discount rate $\gamma$ | 0.995 |
| GAE parameter | 0.97 |
| Minibatches per epoch | 32 |
| Epochs per update | 10 |
| Optimizer | Adam |
| Learning rate $\phi$ | 3e-4 |
| Minimum batch size ($n$) | 2048 |
| $\epsilon$ | 0.2 |
| $\epsilon_1$ | 0.1 |
| weighting parameter $\beta$ | 0.3 |

