# OpenReview forum: "Reflective Policy Optimization"
_ICLR.cc/2024/Conference — Submitted to ICLR 2024_

### Official Review · Reviewer_X1ce · 2023-10-27

**Soundness:** 3 good
**Presentation:** 4 excellent
**Contribution:** 3 good
**Rating:** 8
**Confidence:** 4

**Summary:**

This paper proposed a new policy optimization method named reflective policy optimization based on expanding surrogate function via condition state-action visitation distributions. This paper also provided a clipped surrogate function(following PPO) for efficient calculating.
Compared with current policy optimization methods, this reflective policy optimization performs better on many benchmarks.

**Strengths:**

This paper proposes a novel method with a tighter bound than TRPO. The clipped optimization objective provided by this method is simple and general enough and can be directly applied to many existing methods to replace the PPO objective. It shows improved or comparable performance in the experiments, however again considering the simplicity and generality of the method, this method is valuable.

Overall, the paper is well-written and easy to follow. Adding a detailed expression of the loss function for k=2 before Theorem 4.2 would have made it easier for me to understand.

**Weaknesses:**

It is better to add some experiments on the environments with discrete action space, e.g. MinAtar[1].

The gap between the original reflective policy optimization and the clipped version cannot be ignored.

The method is sensitive to some important hyperparameters, as shown in Figure 4, in Swimmer and Walker2d.


[1] Young, K. Tian, T. (2019). MinAtar: An Atari-Inspired Testbed for Thorough and Reproducible Reinforcement Learning Experiments. arXiv preprint arXiv:1903.03176.

**Questions:**

In the ablation experiment for k, when k increases the variance of the objective function will increase significantly, which puts forward a high requirement for fine adjustment of $\epsilon$ and $\beta$. So I want to know what is the setting of these hyperparameters in the ablation experiment. How to set such hyperparameters to get a reliable result, choosing the right k?

---

> ### Author Response · Authors · 2023-11-19
> **Response to reviewer X1ce**
>
> We are profoundly grateful for the time you have taken from your busy schedule to provide a detailed evaluation of our work. We sincerely appreciate your professional feedback. We are committed to addressing the issues raised and are confident that our revisions will significantly improve the contribution of our work to the RL community.
>
> **Question 1**: In the ablation experiment for k, when k increases the variance of the objective function will increase significantly, which puts forward a high requirement for fine adjustment of $\epsilon$ and $\beta$. So I want to know what is the setting of these hyperparameters in the ablation experiment. How to set such hyperparameters to get a reliable result, choosing the right k?
>
> **Answer 1**: Your point is valid. In the ablation study, as shown in Fig. 5, when k=3, the objective function contains five hyperparameters, which take the values of ($\epsilon=0.2, \epsilon_1=0.1, \beta_1=0.3, \epsilon_2=0.1,\beta_2=0.1$). We fixed $\epsilon, \epsilon_1, \beta_1$ and $\epsilon_2$, and did not extensively fine-tune the hyperparameters, only $\beta_2\in${0.1, 0.3}. Considering the variance factor, we adjust the hyperparameters according to the following rules: $ \epsilon\geq\epsilon_1\geq\epsilon_2\geq \cdots$ , and $ \beta_1\geq\beta_2\geq\cdots$ . In this way, a more reliable result can be obtained. To verify the effectiveness of the  RPO and how to choose k, we conducted experiments on the CliffWalking (see Fig. 2). It was found experimentally that when k = 3, the results obtained were a little better than when k = 2, but the improvement was not significant. Again considering the factor of the number of hyperparameters, we suggest a choice of k of 2.
>
> **Question 2**: (The first point in Weaknesses) It is better to add some experiments on the environments with discrete action space, e.g. MinAtar[1].
>
> **Answer 2**: We sincerely accept your expert guidance. Firstly, we have thoroughly read your reference: Young, K. & Tian, T. (2019). MinAtar: An Atari-Inspired Testbed for Thorough and Reproducible Reinforcement Learning Experiments. arXiv preprint arXiv:1903.03176. We have conducted additional experiments in the discrete action environment. We are immensely grateful for the expert guidance you provided on the MinAtar environment, as well as the relevant literature you recommended. The content of the literature has offered us a substantial opportunity for in-depth learning. Following your suggestions, we have extended our experimental scope to the more challenging Atari environment beyond MinAtar, aiming to demonstrate the comprehensive performance of our RPO algorithm. We look forward to your evaluation of these additional experimental results. We randomly selected twelve Atari environments for these experiments. The results of the MinAtar and Atari experiments have been incorporated into the new version of our paper for your review.
>
> **Question 3**: (The second point in Weaknesses) The gap between the original reflective policy optimization and the clipped version cannot be ignored.
>
> **Answer 3**: We also recognize the phenomenon you pointed out, which is indeed critical and important. However, our experiments show that our clipping operation did not significantly alter the optimal policy but rather made the training more stable during policy updates in most environments.
>
> **Question 4**: (The third point in Weaknesses) The method is sensitive to some important hyperparameters, as shown in Figure 4, in Swimmer and Walker2d.
>
> **Answer 4**: We are deeply appreciative of your thorough and professional insights. Indeed, as you have pointed out, we have observed sensitivity to parameter variations in specific scenarios. Through ablation studies, it gives some insight into fine-tuning the parameters of the algorithm. For some environments, we should reduce the parameter $\beta$ value, reducing the sensitivity of the parameters.
>
>
>
> Once again, thank you for your thorough review and valuable comments on our work. We look forward to your further guidance.

---

> > ### Comment · Reviewer_X1ce · 2023-11-22
> > **appreciate the reply**
> >
> > Thank you for taking the time to answer my questions.
> > The additional experiments conducted by the author have assuaged some of my concerns. From the point of view of the simplicity of the method and empirical performance, I am inclined to maintain a positive perspective in my commentary.

---

> > > ### Author Response · Authors · 2023-11-22
> > > **Official Comment by Authors**
> > >
> > > We deeply appreciate the effort and time you've invested in evaluating our work and discussions, and we are delighted to receive your approval!
> > >
> > > --Best wishes from all the authors.

---

### Official Review · Reviewer_WKkT · 2023-10-30

**Soundness:** 3 good
**Presentation:** 3 good
**Contribution:** 2 fair
**Rating:** 5
**Confidence:** 4

**Summary:**

In this paper, authors propose a Reflective Policy Optimization(RPO) that considers state-action pairs for n-steps in the policy optimization algorithm. The paper theoretically shows that the relationship between the performance of two policies depends on the next state-action pair and performance’s lower bound can be similar to Trust Region Policy Optimization(TRPO). Authors prove that as n increases, the solution space shrinks and can be contained in the solution space of the (n-1) steps. The practical implementation is based on Proximal Policy Optimization(PPO), and experimental results show that it has better convergence speed and average return performance than the baseline algorithm in cliff-walking and mujoco environments.

**Strengths:**

1) For two policies, authors propose to maximize a generalized lower bound that directly takes into account that policy performance is related to the next state-action pair. In particular, they interestingly prove that the optimized policy is reflective through some theory and show that TRPO is a special case of this method, which allows the policy to be monotonically improved.
2) In this paper, authors consider multi-step RL directly from the policy optimization perspective, which is different from the previously proposed multi-step value estimation. In addition, for the practical implementation of the proposed algorithm, the objective function based on PPO was proposed and the correlation between hyperparameters was explained.
3) In the cliff walking environment, the proposed algorithm was experimentally shown to fall off the cliff less and reach the goal faster than the existing algorithms, which is consistent with the performance expected by the authors at the beginning of the paper. It also improved the average return and convergence speed compared to the existing algorithms in the mujoco environment, which is commonly tested in policy optimization.

**Weaknesses:**

1) The theorems and proofs proposed in the paper are interesting, but the performance shown in the experiments does not seem to be much different from the performance of the baselines. In particular, there are many mentions of convergence speed, but the results shown do not show a significant improvement over existing methods. It is necessary to show the performance improvement in an environment where the subsequent state-action can be better considered.
2) The clipping working environment that the authors consider seems a little less relevant to the general field of policy optimization. The proposed environment is more appropriate to be considered in a constrained reinforcement learning or safe reinforcement learning environment where dangerous situations should be avoided. Therefore, the comparison target and baseline algorithm should be the constrained reinforcement learning or safety reinforcement learning algorithm.
3) In the introduction section of the paper, it is argued that existing policy optimization algorithms do not directly consider the impact of subsequent state-actions in the trajectories. Also, in the example, if the agent repeatedly visits the cliff state, it is dangerous because agent is likely to act to fall off the cliff. However, in policy optimization, since there is a discounted sum of reward term, it is possible to provide feedback on the cliff state and the falling action. Therefore, the motivation of the proposed method is difficult to understand, as it allows us to consider the impact of subsequent state action under the influence of rewards.
4) The proposed algorithm increases the number of hyperparameters as the value of k increases. The authors claim that k=2,3 is suitable because of the stability of learning, but even with k=3, it requires 5 hyperparameters to adjust the clipping and learning rate. In addition, the experiments show the case of k=2, and the best performing hyperparameters are not consistent across experimental environments. Therefore, it can be said that the algorithm is sensitive to hyperparameters.

**Questions:**

1) Please answer the questions posed in the weaknesses
2) In Section 4, a new generalized lower bound is defined via Theorem 4.1. In this part, the authors introduce a new surrogate objective function, which has a slightly different semantics than the surrogate objective function in Theorem 3.1, just minus one. Also, they say that they directly optimize the information of current and future state-action pairs, can you explain what the difference is, or can you compare the difference experimentally?
3) The paper states that it applied multi-step RL directly from a policy optimization point of view, but it seems that Generalized Adversarial Estimation (GAE) was also used when training. could you tell me if there was any correlation between the GAE parameters and multi-step k in experiments?
4) The paper shows that the convergence speed increases with the value of K, but the performance is only shown in three environments. Can you explain whether the performance varies significantly depending on the difficulty of the environment or the dimensionality of the state and action?

---

> ### Author Response · Authors · 2023-11-19
> **Response to reviewer WKkT (Part 1/2)**
>
> Thank you very much for your insightful feedback. Receiving such professional guidance from you is indeed a privilege for us. We are committed to addressing the issues raised and are confident that our revisions will significantly improve the contribution of our work to the RL community. We sincerely request that you reconsider your evaluation of our manuscript and consider revising your score accordingly. Your continued support and feedback are invaluable to us.
>
> **Question 1**: (Weaknesses 1) The performance shown in the experiments does not seem to be much different from the performance of the baselines.
>
> **Answer 1**: We are grateful for your query about the performance improvement and apologize for not articulating this clearly in our paper. In our experiments, we found that our method has a significant advantage over the original PPO. However, to showcase the robustness and broad applicability of our approach, we did not perform specific parameter tuning. The results post-tuning will be detailed in the paper's appendix to transparently exhibit RPO’s performance. In Fig. 3, our method is compared not only with the PPO algorithm but also with some algorithms that have modified PPO from an on-policy to an off-policy approach (GePPO and OTRPO) and it is found that our method is also advantageous. Figure 1 shows that the direct use of subsequent data plays a positive role and our approach can be seamlessly integrated into existing algorithm frameworks.
>
> We once again apologize for any confusion caused by our lack of clarity in conveying that specific parameter tuning was not conducted to demonstrate the robustness and general adaptability of our method. Please forgive any inconvenience this may have caused during your review. Your guidance is invaluable, and we will take greater care in this regard in our subsequent paper submissions.
>
> **Question 2**: (Weaknesses 2) The proposed environment is more appropriate to be considered in a constrained RL or safe RL.
>
> **Answer 2**: Thank you for your approval of our proposed algorithm. Our proposed method serves as a supplement to existing reinforcement learning approaches. It is not only applicable to classic reinforcement learning scenarios but may also play a more significant role in constrained reinforcement learning or safe reinforcement learning.  It also requires further research on this algorithm in order to better apply it in different environments. It will be a point that can be studied in the future.
>
> **Question 3**: (Weaknesses 3) About the motivation of the proposed method.
>
> **Answer 3**: I am very sorry that we did not make this example more straightforward. We will modify it in the introduction. With this example, we want to consider the impact of the subsequent data directly. The value function potentially contains information about the subsequent data. We need to consider the question: Is it the best way to optimize a policy using only value functions? The answer is definitely not. There is another way to optimize the current policy by directly using the subsequent data, unlike optimizing the policy using only the value function. Intuitively, the direct use of the subsequent data may speed up the algorithm's convergence  and improve sample efficiency. In the subsequent sections of this paper, we verify this intuition theoretically and experimentally, respectively.
>
> **Question 4**: (Weaknesses 4) Sensitivity of hyperparameters.
>
> **Answer 4**: Fine-tuning the hyperparameters of our algorithm indeed leads to improved performance. In the main experiment (as shown in Figure 3), we fixed the hyperparameters without fine-tuning  for a fair comparison. From Fig. 3, even with fixed hyperparameters, our approach achieves good results compared with PPO and the off-policy version of PPO.
>
> **Question 5**: (Original question 2) About theorem 3.1 and 4.1, can you explain what the difference is, or can you compare the difference experimentally?
>
> **Answer 5**: Consider $L_1(\pi, \hat{\pi})$ of the theorem 3.1 as an example. If the environment is unknown, it can only be optimized by sampling. If we sample a trajectory $\tau$, and $(s_0, a_0, s_1, a_1)\in\tau$, if $ A^{\hat{\pi}}(s_1, a_1)<0 $ and $ r_0-1 <0$, we know that $ (r_0-1)r_1 A^{\hat{\pi}}(s_1, a_1)>0 $. Optimizing the $ L_1(\pi, \hat{\pi}) $ (considering the extreme case, using one sample), the probability of $ a_1 $ is increased. However, $ A^{\hat{\pi}}(s_1, a_1)<0 $, we should decrease the probability of $ a_1 $. It's a contradiction. Thus this term "1" of $ r_0-1 $ may adversely affect policy optimization though the theory is sound. This situation exists when the environment is unknown. But for $\hat{L}_1(\pi, \hat{\pi})$ in the theorem 4.1, there is no such problem. We also experimentally compared these two methods, *i.e.*, RPO vs.* TayPO (with the term ”1"). The possibility of what we said earlier is also verified from the experiment.

---

> ### Author Response · Authors · 2023-11-19
> **Response to reviewer WKkT (Part 2/2)**
>
> **Question 6**: (Original question 3) Could you tell me if there was any correlation between the GAE parameters and multi-step k in experiments?
>
> **Answer 6**: In this paper, GAE is evaluated in the same way as PPO, *i.e.*, $\hat{A}^{GAE}_t=\delta^t+(\gamma \lambda)\delta^{t+1}+\cdots+(\gamma \lambda)^{T-t+1} \delta^{T-1}  $,
>
>  where $ \delta^t=r_t+\gamma V(s_{t+1})-V(s_t)$.
>
> Assume that the sampled trajectory is $\tau = (s_0,a_0,s_1,a_1, \cdots, s_t, a_t, \cdots)$, we can calculate the advantage functions ($\hat{A}^{GAE}_0, \hat{A}^{GAE}_1, \cdots, \hat{A}^{GAE}_t, \cdots$).
>
> When k=2, if we sample ($s_t, a_t, s_{t+1}, a_{t+1}$), the current policy is optimized by ($\hat{A}^{GAE}_t$,
>
> $\hat{A}^{GAE}_{t+1}$). It can be seen that the relationship between them is that when k is fixed, the number of advantage functions is also fixed.
>
> **Question 7**: (Original question 4)   The performance is only shown in three environments about with the value of k.
>
> **Answer 7**: Previously, due to limited resources and time, we did not perform parameter tuning for k=3. We only ran some experiments in three environments and included these results in our paper. Now, we have supplemented this experiment in the Appendix.

---

> > ### Comment · Reviewer_WKkT · 2023-11-21
> >
> > To the author,
> >
> > Thank you for taking the time to address the shortcomings and questions raised. This helped me to understand the paper.
> >
> > Sincerely.

---

> > > ### Author Response · Authors · 2023-11-23
> > > **Have we addressed your concerns?**
> > >
> > > Thanks again for your time and effort in reviewing our paper! As the discussion period is coming to a close, we would like to know if we have resolved your concerns expressed in the original reviews. We remain open to any further feedback and are committed to making additional improvements if needed. If you find that these concerns have been resolved, we would be grateful if you would consider reflecting this in your rating of our paper.

---

> ### Comment · Area_Chair_2DfF · 2023-11-20
>
> Dear WKkT,
>
> The author reviewer discussion period is ending soon this Wed. Does the author response clarify your concerns w.r.t., e.g., the empirical aspects of this work, or there are still outstanding items that you would like to see more discussion?
>
> Thanks again for your service to the community.
>
> Best,
> AC

---

> ### Author Response · Authors · 2023-11-21
> **Some additional supplementary experiments.**
>
> Thanks again for your time and effort in reviewing our paper! To further allay your concerns, we add some Atari experiments. We remain open to any further feedback and are committed to making additional improvements if needed.
>
> Table 1 Comparison of RPO and PPO performance at 60% timesteps in MuJoCo.
> | Environments |  Timesteps   |    RPO     |   PPO    |
> | :----------: | :---------: | :--------: | :------: |
> | HalfCheetah  | 0.6 million | **3100.0** |  2190.4  |
> |   Reacher    | 0.6 million |  **-5.7**  | **-5.7** |
> |   Swimmer    | 0.6 million | **155.1**  |  133.9   |
> |   Walker2D   | 0.6 million | **3855.8** |  3402.6  |
> |    Hopper    | 0.6 million | **3206.0** |  2720.3  |
> |   Humanoid   | 3.0 million | **5349.3** |  4835.4  |
>
> Table 2 Comparison of RPO and PPO performance at 60% timesteps in Atari.
> | Environments |  Timesteps  |     RPO     |    PPO    |
> | :----------: | :--------: | :---------: | :-------: |
> |   Asterix    | 30 million | **9879.4**  |  8346.6   |
> |  BattleZone  | 30 million | **29640.7** |  21862.5  |
> |    Boxing    | 30 million |  **98.3**   |   87.5    |
> |   Breakout   | 30 million |  **300.8**  |   200.4   |
> |  Centipede   | 30 million | **5752.2**  |  3246.5   |
> |  DoubleDunk  | 30 million |  **-5.3**   |   -7.7    |
> |    Enduro    | 30 million |    311.5    | **371.3** |
> | FishingDerby | 30 million |  **30.2**   |   24.6    |
> |   Kangaroo   | 30 million | **9974.6**  |  8907.4   |
> |   Phoenix    | 30 million | **28538.1** |  20409.7  |
> |    Pong    | 30 million |  **21.0**   |   **21.0**    |
> |    Qbert     | 30 million | **14308.4** |  13933.0  |
>
> In order to better present the advantages of this method RPO over PPO, in addition to our experiments on MuJoCo, we randomly selected 12 Atari environments for our experiments (runs 50 million, 3 seeds).
> As shown in Tables 1 and 2, our proposed algorithm RPO achieves better results compared to PPO at the 60% timesteps in MuJoCo and Atari.  For more information, please see the appendix.

---

### Official Review · Reviewer_KkoJ · 2023-10-31

**Soundness:** 1 poor
**Presentation:** 2 fair
**Contribution:** 1 poor
**Rating:** 3
**Confidence:** 5

**Summary:**

The paper considers policy optimization from the trajectory perspective and proposes to optimize polices with the information from both the current state-action and the subsequent state-action pairs. To achieve this, the paper unfolds the performance improvement lemma over time steps to get the unrolled surrogate objective, i.e., the generalized surrogate, and a “residual” term. The paper then shows the “residual term” can be well bounded, yielding a policy improvement strategy: by iteratively pushing up the lower bound. The paper then use the same clipping scheme from the PPO and forms the Reflective Policy Optimization (RPO). This method has been evaluated on Mujoco control benchmarks and showed good performance.

**Strengths:**

### Originality
The idea of consider the subsequent state-action pairs in the policy optimization sounds interesting and novel. The theory (if correct) can add new insights into the policy gradient methods when considering unrolling it for a long horizon.

**Weaknesses:**

### Quality & significance
**There can be some technical misstatement & errors in the main contributions**.

First, the following statement can be erroneous: “If you recombine the above equation $(r_0-1)r_1 A^{\hat{\pi}}(s_1, a_1)]\cdot r_1 >0$, optimizing it will be found to increase the probability of $a_1$ . However, $A_{\hat{\pi}}(s_1 , a_1) < 0$, we should decrease the probability of $a_1$ . This would present a contradiction.” **Optimizing this objective can result in a bit more complicated consequence than what’s been stated here**: since the policy in both $r_0$ and $r_1$ is parameterized with a same set of parameters, optimizing the objective may increase the probability of $a_0$ and thus tip over the sign of $r_0-1$.  The problem is $a_0$ and $a_1$ are from the same parametrized policy.


Second, **the monotonic improvement statement of Theorem 4.1 can be erroneous**. Consider a special case of $k=1$, which yields the TRPO bound as suggested by the paper (assume this statement is correct), then the lower bound will be $\alpha_0 \hat{L}_0(\pi, \hat{\pi}) - \hat{C}_1(\pi, \hat{\pi})$,

which equals to $E_{s_0, a_0\sim\rho^{\hat{\pi}}} [r_0 A^{\hat{\pi}}(s_0, a_0)] - \frac{\gamma R_{max}}{(1-\gamma)^3}||\pi - \hat{\pi}||_1^2$. This is the surrogate objective implied by Theorem 4.1 and should be improved upon. However, **this surrogate objective is strictly negative for $\gamma=0.995$ used by this paper**: consider the first term

$E_{s_0, a_0\sim\rho^{\hat{\pi}}} [r_0 A^{\hat{\pi}}(s_0, a_0)] = E_{s_0, a_0\sim\rho^{\hat{\pi}}} [ (r_0-1) A^{\hat{\pi}}(s_0, a_0)] $ (this is because the advantage is defined on $\hat{\pi}$ and the integral over $\hat{\pi}$ will be zero. Hence,

$E_{s_0, a_0\sim\rho^{\hat{\pi}}} [ (r_0-1) A^{\hat{\pi}}(s_0, a_0)] $

$\leq \frac{  ||\pi - \hat{\pi}||_1^2 Rmax }{1-\gamma} $ (this is a direct result from the proof of Corollary A.2 in appendix, i.e., $G_1$).

Thus,

$E_{s_0, a_0\sim\rho^{\hat{\pi}}} [r_0 A^{\hat{\pi}}(s_0, a_0)] - \frac{\gamma R_{max}}{(1-\gamma)^3}||\pi - \hat{\pi}||_1^2$

$\leq \frac{  ||\pi - \hat{\pi}||_1^2 Rmax }{1-\gamma} - \frac{\gamma Rmax}{(1-\gamma)^3}||\pi - \hat{\pi}||_1^2$

$\leq \frac{  ||\pi - \hat{\pi}||_1^2 Rmax }{1-\gamma} [ 1- \frac{\gamma}{(1-\gamma)^2}] < 0$ (for $\gamma =0.995)$

So, the maximum value of the lower bound is strictly negative. **It turns out to be impossible to get "a monotonically improving sequence of policies $\{\pi_i\}_{i=1}^{\infty}$ satisfying $\eta(\pi_0) \leq \eta(\pi_1) \leq ...$" for this case $k=1$**.

Please correct me if my understanding is wrong.

### Clarity
There are some confusing points and unclear definitions. See in my questions.

**Questions:**

1. I don’t see the point in the examples of “cliff” and “treasure” states in the second paragraph of Introduction Section. Would the value function just suffice whether a state is favourable?

2. Most of the citations are using a wrong format. It seems that the author misused the citep & citet.

3. there is a notation issue in the second line of $\eta(\pi) - \eta(\hat{\pi})$. The first term on RHS should be with (s_0, a_0).

4. The $||\pi - \hat{\pi}||_1$ has not been defined in the paper.

5. The definition of the conditional occupancy measure is a bit confusing. From the proofs in the appendix, this conditional occupancy measure is a distribution with the conditions on the initial state-action distribution, i.e., the stationary state distribution induced from a predefined initial state distribution. Then I’m not sure if $(s_0, a_0)\sim\rho$ and $(s_i, a_i)\sim\rho(\cdot|s_{i-1}, a_{i-1})$ can be defined coherently.

Some language issues (those do not affect my assessment)
* “between the performance of \pi and \hat{\pi} from a trajectory-based.”

---

> ### Author Response · Authors · 2023-11-19
> **Response to reviewer KkoJ  (Part 1/2)**
>
> Thank you very much for your insightful feedback. We greatly appreciate your constructive suggestions, which have provided valuable guidance for enhancing the quality of our manuscript. We are committed to addressing the issues raised and are confident that our revisions will significantly improve the contribution of our work to the RL community. We sincerely request that you reconsider your evaluation of our manuscript and consider revising your score accordingly. Your continued support and feedback are invaluable to us.
>
> **Question 1**: About the examples of “cliff” and “treasure” stated in the second paragraph of the Introduction Section. Would the value function suffice whether a state is favorable?
>
> **Answer 1**: I am very sorry for not making this example clear. We will modify it in the introduction. Let's briefly restate the example. Consider an environment with a "cliff." what would an agent do if it performed an action under a state and fell into a "cliff"? This action is dangerous. Meanwhile, this state might also be dangerous due to the fact that the next time the agent reaches this state again, it is likely to perform the same action. Hence, the agent also needs to avoid getting to this state again and keep out of this state as much as possible. The previous action, when reaching this state, the previous action also needs to directly avoid being performed because it is possible to fall into that state again. The same result is found for the "treasure" environment.  Hence, it is necessary to optimize the previous action directly with the subsequent state-action pairs information, not only through the value function, thereby improving sample efficiency.
>
> Yes, this way is feasible if the policy is optimized using the value function of the current state due to the fact that the value function potentially contains information about the subsequent states. We need to consider the question: Is it the best way to optimize a policy using only value functions? The answer is definitely not. What we are trying to convey with this example is that there is another, better way to optimize the current policy by directly using the subsequent data, unlike optimizing the policy using only the value function. Intuitively, the direct use of the subsequent data may speed up the convergence of the algorithm and improve sample efficiency. In the subsequent sections of this paper, we verify this intuition theoretically and experimentally, respectively.
>
> **Question 2** : About the citep \& citet.
>
> **Answer 2**: Thank you very much for pointing this out, we revise it in the next version.
>
> **Question 3** : In the second line of $\eta(\pi)-\eta(\hat{\pi})$, the first term on RHS should be with ($s_0, a_0$).
>
> **Answer 3**: Thank you very much, we revise it in the next version.
>
> **Question 4** : The $\|\pi-\hat{\pi}\|_1 $ has not been defined in the paper.
>
> **Answer 4**: Thank you very much, we revise it. $\|\pi-\hat{\pi}\|_1$  is defined as $\max_s\sum_a|\pi(a|s)-\hat{\pi}(a|s)|$.
>
> **Question 5** : About $ (s_0, a_0)\sim \rho $, and $ (s_i, a_i)\sim \rho(\cdot|s_{i-1},a_{i-1}) $.
>
> **Answer 5**: I'm very sorry, I didn't make that clear. Let us explain in detail below. From the definition of $ \rho(s) $ in Preliminaries Section, we see that $ \mathbb{P}(s_t=s|\rho_0, \pi) $ has Markov properties. So, $ \rho^{\pi}(s) $ has the same properties. From the proof of Theorem A.1 (see appendix), the link between the performance difference of $ \pi $ and $ \hat{\pi} $ and state visit distribution and conditional state visit distribution has been established. We know that $ \rho(s_i|s_{i-1}, a_{i-1})= \rho(\cdot|s_{i-1}, a_{i-1}, s_{i-2}, a_{i-2}, \cdots, s_0, a_0)$, which is the conditional distribution under $ s_{i-1} $ and $ a_{i-1} $, and $ s_{i-1}\sim\rho(\cdot|s_{i-2}, a_{i-2}) $ and $ a_{i-1}\sim \pi(\cdot|s_{i-1}) $. For example, given the distributions of $ \rho(s_0) $ and $ \pi(a_0|s_0) $, $ \rho(s_1|s_{0}, a_{0}) $ must exist according to Bayesian theory, but the exact form is difficult to give directly. Their relationship is $ \rho(\cdot) = \int \rho(\cdot|s_0, a_0) \rho(s_0) \pi(a_0|s_0) ds_0da_0 $. For the general case, we have $\rho(\cdot) = \int \rho(\cdot|s_{i-1}, a_{i-1})\rho(s_{i-1}|s_{i-2}, a_{i-2})\pi(a_{i-1}|s_{i-1})\cdots \rho(s_0) \pi(a_0|s_0) ds_{i-1}da_{i-1}\cdots ds_0da_0 $.

---

> ### Author Response · Authors · 2023-11-19
> **Response to reviewer KkoJ  (Part 2/2)**
>
> **Question 6** : (Weaknesses) about $ (r_0-1)r_1 A^{\hat{\pi}}(s_1, a_1) = [(r_0-1) A^{\hat{\pi}}(s_1, a_1)]\cdot r_1>0 $ ... present a contradiction. Optimizing this objective can result in a bit more complicated consequence than what’s been stated here.
>
> **Answer 6**: Your point is valid. From the perspective of parameter optimization, optimizing this objective be considered a more complex situation. In this paper, we consider this function intuitively without focusing on the specific form of the parameters. When the model is unknown, we optimize this objective function by sampling and give intuitively the possible problems which there is no way to avoid.
>
> **Question 7** : (Weaknesses) about the monotonic improvement statement of Theorem 4.1.
>
> **Answer 7**: Thank you for providing a proof, the approach of proof is correct. We have identified a flaw in one of the proof steps, *i.e.*, it is not true that $ E_{s_0, a_0\sim\rho^{\hat{\pi}}}[(r_0-1) A^{\hat{\pi}}(s_0, a_0)]\frac{\|\pi-\hat{\pi}\|_1^2R\max}{1-\gamma}$,
>
> but rather that $ E_{s_0, a_0 \sim \rho^{\hat{\pi}}}[(r_0-1) A^{\hat{\pi}}(s_0, a_0)] \leq \frac{\|\pi-\hat{\pi}\|_1 R\max}{1-\gamma}$.
>
> We can see that $ G_1(\pi, \hat{\pi}) = E_{s_0, a_0 \sim \rho^{\hat{\pi}}(\cdot)}(r_0-1) E_{s_{1}, a_{1} \sim \rho^{\pi}(\cdot|s_{0}, a_{0})}A^{\hat{\pi}}(s_1, a_1) $ is different from $ E_{s_0, a_0 \sim \rho^{\hat{\pi}}}[(r_0-1) A^{\hat{\pi}}(s_0, a_0)]$.
>
> Therefore, the results of $ G_1 $ cannot be used directly. We give the complete proof below.
> $ E_{s_0, a_0 \sim \rho^{\hat{\pi}}}[(r_0-1) A^{\hat{\pi}}(s_0, a_0)] - \frac{ \gamma R\max}{(1-\gamma)^2}\|\pi-\hat{\pi}\|_1^2\leq \frac{ R\max}{1-\gamma}\|\pi-\hat{\pi}\|_1 -\frac{ \gamma R\max}{(1-\gamma)^2}\|\pi-\hat{\pi}\|_1^2 $.
>
> When $ 0<\|\pi-\hat{\pi}\|_1<\frac{1-\gamma}{\gamma} $, the upper bound is greater than zero. So the lower bound of performance has solutions.

---

> > ### Comment · Reviewer_KkoJ · 2023-11-21
> > **Post-rebuttal**
> >
> > Re the answer to Q6, I don't think the authors address my concern. This is actually a crucial motivating example to introduce the "reflective" idea. A rigorous and flawless descriptions should be given.
> >
> > Re the answer to Q7, I think my confusion point was the definition of $\rho(\cdot|s_{i-1}, a_{i-1})$ (thank you for clarifying this but I'm still not confident in this definition). Anyway, the authors' new derivation appears to be wrong to me. Specifically, given the authors' bound on the first term, I update the bound as follows
> >
> > $E_{s_0, a_0\sim\rho^{\hat{\pi}}} [r_0 A^{\hat{\pi}}(s_0, a_0)] - \frac{\gamma R_{max}}{(1-\gamma)^3}|\pi - \hat{\pi}|_1^2$
> >
> > $\leq \frac{  |\pi - \hat{\pi}|_1 Rmax }{1-\gamma} - \frac{\gamma Rmax}{(1-\gamma)^3}|\pi - \hat{\pi}|_1^2$
> >
> > $= \frac{Rmax}{1-\gamma}( |\pi - \hat{\pi}|_1 - \frac{\gamma}{(1-\gamma)^2}|\pi - \hat{\pi}|_1^2)$
> >
> > This quadratic form takes the maximal when $|\pi - \hat{\pi}|_1 = \frac{(1-\gamma)^2}{2\gamma}\approx 1e^{-5}$ for $\gamma=0.995$. Further note that $|\pi - \hat{\pi}|_1 = \max_s \sum_a|\pi(a|s) - \hat{\pi}(a|s)|$. Not sure how the policy can be updated from $\hat{\pi}$ to $\pi$ given that the maximum update over all state space is bound by an infinitesimal value $1e^{-5}$.

---

> ### Comment · Area_Chair_2DfF · 2023-11-20
>
> Dear KkoJ,
>
> The author reviewer discussion period is ending soon this Wed. Does the author response clarify your concerns about technical correctness and presentation, or there are still outstanding items that you would like to see more discussion?
>
> Thanks again for your service to the community.
>
> Best,
> AC

---

> ### Author Response · Authors · 2023-11-22
> **Response to Post-rebuttal**
>
> Thank you for your feedback. We greatly appreciate your constructive suggestions, which have provided valuable guidance for enhancing the quality of our manuscript. We will answer your questions as best we can. Lastly, we thank you for taking the time to engage with our work.
>
> **Re-Question 6**:  Rigorous and flawless descriptions should be given as an example.
>
> **Re-Answer 6**:  I am very sorry for not making this example clear. I've added a more rigorous description in our manuscript, and we've explained it to you to make it easier for you to understand.
>
> Consider $L_1(\pi, \hat{\pi})$ in Eqn. (2) as an example.  We consider this function without focusing on the specific form of the parameters. When the environment is unknown, it can only be optimized by sampling. Considering the extreme case, the function  $L_1(\pi, \hat{\pi})$ is optimized by using a sample $(s_0, a_0, s_1, a_1)$, *i.e.*, $L_1(\pi, \hat{\pi})\approx(r_0-1)r_1 A^{\hat{\pi}}(s_1, a_1) $.
>
> If $ A^{\hat{\pi}}(s_1, a_1)<0 $ and $ r_0-1 <0$, we know that $ (r_0-1)r_1 A^{\hat{\pi}}(s_1, a_1)=[(r_0-1)A^{\hat{\pi}}(s_1, a_1)]r_1>0 $. The probability of $ a_1 $ is increased. However, when $ A^{\hat{\pi}}(s_1, a_1)<0 $, we should  decrease the probability of $ a_1 $. It's a contradiction. Thus, this term "1" of $ r_0-1 $ may adversely affect policy optimization, though the theory is sound. This situation exists when the environment is unknown.
>
> **Re-Question 7** : (Weaknesses) About the monotonic improvement statement of Theorem 4.1.
>
> **Re-Answer 7**:  Firstly, we define $\rho(\cdot|s, a)$ from another perspective.
>
> Inspired by discount state visitation distribution $\rho(\cdot)$, *i.e.*,
> $$\rho^{\pi}(s) = (1-\gamma)\sum_{t=0}^{\infty}\gamma^t\mathbb{P}(s_t=s|\rho_0, \pi),$$
> we can similarly  give a definition of $\rho^{\pi}(s,a,s')$. It is defined as $$\rho^{\pi}(s,a,s') = (1-\gamma)\sum_{t=0}^{\infty}\gamma^t\mathbb{P}(s_t=s,a_t=a,s_{t+1}=s'|\rho_0, \pi),$$
> where $\mathbb{P}(s_t=s,a_t=a,s_{t+1}=s'|\rho_0, \pi)=\mathbb{P}(s_t=s|\rho_0, \pi)\pi(a_t=a|s_t=s)P(s_{t+1}=s'|s_t=s,a_t=a),$ Further, we can give the definition of $\rho^{\pi}(\cdot|s, a)$, *i.e.*, $\rho^{\pi}(s'|s, a)=\frac{\rho^{\pi}(s, a,s')}{\rho^{\pi}(s, a)}$.
>
> By this way, $\rho^{\pi}(s_2|s_1, a_1)=\frac{\rho^{\pi}(s_0,a_0,s_1,a_1,s_2)}{\rho^{\pi}(s_1,a_1|s_0,a_0)\rho^{\pi}(s_0,a_0)}$. Similarly, we can define $\rho^{\pi}(s_i|s_{i-1}, a_{i-1})$.
>
> Secondly, thank you for re-providing proof. The approach of proof is correct. Moreover, we have identified a flaw in one of the proof steps. The penalty term in the lower bound is $\frac{ \gamma R\max }{(1-\gamma)^2}\|\pi-\hat{\pi}\|_1^2$ instead of $\frac{ \gamma R\max }{(1-\gamma)^3}\|\pi-\hat{\pi}\|_1^2$ (see theorem 4.1 of my paper or theorem 1 of the TRPO paper [1]). So, the quadratic form takes the maximal when $\|\pi-\hat{\pi}\|_1=\frac{1-\gamma}{2\gamma}$.
>
> Lastly, we are well aware of your concern that $\|\pi-\hat{\pi}\|_1$ may be too small. By the analysis of the theorem, we are more interested in the existence of solutions. In practice, as the TRPO paper says, "the KL divergence is bounded at every point in the state space, ...,  is impractical to solve due to the large number of constraint".  This issue is familiar to our approach but rather a challenge universally encountered by all lower bound algorithms, *e.g.*, TRPO [1], CPO [2], GePPO [3], and so on.
>
> [1] Schulman J, Levine S, Moritz P,et al. Trust Region Policy Optimization. ICML, 2015.
>
> [2] Achiam J , Held D, Tamar A,et al. Constrained Policy Optimization. ICML, 2017.
>
> [3] Queeney J , Paschalidis I C,et al. Generalized Proximal Policy Optimization with Sample Reuse. NeurIPS, 2021.

---

> > ### Comment · Reviewer_KkoJ · 2023-11-22
> > **penalty term**
> >
> > Why the penalty term is $\frac{\gamma Rmax}{(1-\gamma)^2}|\pi - \hat{\pi}|_1^2$? Based on the Theorem 4.1, if we consider the case $k=1$, the penalty term should be $\frac{\gamma Rmax}{(1-\gamma)^3}|\pi - \hat{\pi}|_1^2$. No?

---

> > > ### Author Response · Authors · 2023-11-23
> > > **Have we addressed your concerns?**
> > >
> > > Thanks again for your time and effort in reviewing our paper! As the discussion period is coming to a close, we would like to know if we have resolved your concerns expressed in the original reviews. We remain open to any further feedback and are committed to making additional improvements if needed. If you find that these concerns have been resolved, we would be grateful if you would consider reflecting this in your rating of our paper.

---

> ### Author Response · Authors · 2023-11-22
> **Response to penalty term**
>
> Thank you for your feedback. We would be honored to respond to each of your queries in detail. Lastly, we thank you for taking the time to engage with our work.
>
> From theorem 4.1, when $k=1$, we can obtain the following inequality
>
> $$\eta(\pi)-\eta(\hat{\pi})\geq \alpha_0 E_{s_0, a_0 \sim \rho^{\hat{\pi}}}[r_0 A^{\hat{\pi}}(s_0, a_0)]- \hat{C}_1(\pi, \hat{\pi}),$$
>
> where $\alpha_0=\frac{1}{1-\gamma}$ and $\hat{C}_1(\pi, \hat{\pi})=\frac{\gamma R\max}{(1-\gamma)^3}\|\pi-\hat{\pi}\|_1^2$.
>
> If the right-hand side of the above inequality is optimized as a whole, the penalty term is $\frac{\gamma R\max}{(1-\gamma)^3}\|\pi-\hat{\pi}\|_1^2$.
>
> If we omit $\alpha_0$, the objective function we optimize is given by
>
> $$E_{s_0, a_0 \sim \rho^{\hat{\pi}}}[r_0 A^{\hat{\pi}}(s_0, a_0)]- \frac{\gamma R\max}{(1-\gamma)^2}\|\pi-\hat{\pi}\|_1^2,$$
>
> so, the penalty term is $\frac{\gamma R\max}{(1-\gamma)^2}\|\pi-\hat{\pi}\|_1^2$.

---

### Official Review · Reviewer_kAky · 2023-10-31

**Soundness:** 2 fair
**Presentation:** 1 poor
**Contribution:** 2 fair
**Rating:** 3
**Confidence:** 2

**Summary:**

The paper proposes a method called Reflexive Policy Optimization, which the authors claim uses information “from trajectory data” more efficiently to optimize a policy. The authors claim that agents have “introspection” and “reflect on prior experience”. The method is claimed to have guaranteed monotonic progress improvement on the original policy performance objective and that it “contracts the solution space of the optimized policy”, thus “expediting” the training procedure. Moreover, the authors claim the method is superior to baselines on Mujoco tasks.

I didn’t really understand the motivation for the paper, and I found it very confusing and difficult to follow. I expand in the following sections.

**Strengths:**

I believe the authors want to say that their method performs a kind of hindsight credit assignment \citep{harutyunyan2019}, but this is just a hunch. I didn't really understand from the text.
The paper appears to be backed by theory and support their superior performance claims with empirical results. Unfortunately, I did not really understand the motivation and method to assess these  properly. Please see below my points of confusion. I am happy to revise my score if the authors the motivation and goal of the paper, and expand on the points below.


@article{harutyunyan2019,
  author       = {Anna Harutyunyan and
                  Will Dabney and
                  Thomas Mesnard and
                  Mohammad Gheshlaghi Azar and
                  Bilal Piot and
                  Nicolas Heess and
                  Hado van Hasselt and
                  Greg Wayne and
                  Satinder Singh and
                  Doina Precup and
                  R{\'{e}}mi Munos},
  title        = {Hindsight Credit Assignment},
  journal      = {CoRR},
  volume       = {abs/1912.02503},
  year         = {2019},
  url          = {http://arxiv.org/abs/1912.02503},
  eprinttype    = {arXiv},
  eprint       = {1912.02503},
  timestamp    = {Wed, 20 Apr 2022 07:47:18 +0200},
  biburl       = {https://dblp.org/rec/journals/corr/abs-1912-02503.bib},
  bibsource    = {dblp computer science bibliography, https://dblp.org}
}

**Weaknesses:**

First, I do not understand why the authors believe proximal methods do not already account for the true gradient of the policy which also considers the contribution through the stationary distribution.  Indeed, in the original CPI paper \citep{kakade}, the authors first derive the form of the bound that contains the stationary distribution with the current policy, and not a prior policy, then use a mixture policy to ensure that the new policy is “close” to the prior one and justify replacing the current stationary distribution with the previous one. However, a lot of work has been done since, and it is known that proximal PG methods \citep{bhandari21, shani2019, vaswani2021} (and algorithmic implementation \citep{li2022analytical}, PPO, TRPO, MDPO \citep{tomar2020}, MPO \citet{abdolmaleki2018} etc.) are functional-gradient methods \citep{vaswani2021}, in the sense that the lower bound the algorithms optimize are linearizations of the policy performance objective in the direct policy representation. The functional gradient is $d_{\pi_t}^\top Q_{\pi_t}$. In that sense the stationary distribution that the algorithms use is w.r.t. (with respect to) the previous policy (see \citet{bandhari2019, vaswani2021, sutton2000, agarwal2019}). As they follow the true policy gradient, these methods are convergent \citep{bhandari21, vaswani2021, xiao2022, johnson2023}, at least their theoretical versions, under adaptive step sizes. The true policy gradient does take into account the gradient through the stationary distribution.
I did not understand at all the examples they give, which seem to say that “0” has some special meaning when it comes to the reward.

The text is very unclear, vague, and grammatically incorrect, which makes it very difficult to follow the authors’ arguments.
In the motivating introduction, the authors discuss multi-step methods, but this is completely out of context, and use words which have not been defined and are not scientifically rigorous, like “we give a nice theory.”, “the policy can be reflective”, “ this empowers the agent to engage in introspection and introduce modifications to its actions within the current state to a certain degree”, “some admirable algorithms”, “direct optimization of this generalized surrogate objective function may have to be done very carefully”, “RPO efficiently utilizes ‘good’ experiences, makes adjustments based on ‘bad’ experiences’”

With respect to scientific correctness, at some point the authors completely remove a term from the bound saying that “this term “1” may adversely affect policy optimization. I did not understand that at all. The authors say “for $k = 1$, the $l_1$ norm constraints are replaced by KL constraints”, but this to me makes no sense at all.

The notation is also confusing, as it seems the authors reuse “r” to represent the importance sampling ratio between the current policy and a prior one. It is unclear what “k” represents as it is not defined, just implied, and seems very important throughout the paper, with multiple reference and impacting the algorithms’ empirical performance.

Then, I also did not really understand the emphasis on the exploration-exploitation coupling on on-policy methods since the whole point of the paper is to improve off-policy methods like PPO and TRPO. These proximal methods are off-policy algorithms, which is the entire point of the efficiency of the methods, the fact that it can reuse prior experience (from the previous policy) multiple times to minimize the lower bound. Otherwise, on-policy policy gradient methods like \citet{sutton2000} (or algorithmic implementations with parallel streams of experiences like A3C \citep{mnih16}, Impala \citep{espeholt18}) directly use the PGT \citep{sutton2000} w.r.t the parameter vector and rebuild the linear lower bound at each time-step so they don’t need a lower bound surrogate objective that is quasi-concave and can be optimized for multiple updates.


The authors claim the method maintains mototonic improvement, but it is known that proximal-gradient methods, with adaptive step size have this property \citep{alfano2023, chen2022, xiao2022, johnson2023}, since they are functional-gradient methods, at least for tabular direct or softmax parametrizations.

@misc{li2022analytical,
      title={An Analytical Update Rule for General Policy Optimization},
      author={Hepeng Li and Nicholas Clavette and Haibo He},
      year={2022},
      eprint={2112.02045},
      archivePrefix={arXiv},
      primaryClass={cs.AI}
}

@InProceedings{bhandari21,
  title = 	 { On the Linear Convergence of Policy Gradient Methods for Finite MDPs },
  author =       {Bhandari, Jalaj and Russo, Daniel},
  booktitle = 	 {Proceedings of The 24th International Conference on Artificial Intelligence and Statistics},
  pages = 	 {2386--2394},
  year = 	 {2021},
  editor = 	 {Banerjee, Arindam and Fukumizu, Kenji},
  volume = 	 {130},
  series = 	 {Proceedings of Machine Learning Research},
  month = 	 {13--15 Apr},
  publisher =    {PMLR},
  pdf = 	 {http://proceedings.mlr.press/v130/bhandari21a/bhandari21a.pdf},
  url = 	 {https://proceedings.mlr.press/v130/bhandari21a.html},
  abstract = 	 { We revisit the finite time analysis of policy gradient methods in the one of the simplest settings: finite state and action MDPs with a policy class consisting of all stochastic policies and with exact gradient evaluations. There has been some recent work viewing this setting as an instance of smooth non-linear optimization problems, to show sub-linear convergence rates with small step-sizes. Here, we take a completely different perspective based on illuminating connections with policy iteration, to show how many variants of policy gradient algorithms succeed with large step-sizes and attain a linear rate of convergence. }
}
@article{mnih16,
  author       = {Volodymyr Mnih and
                  Adri{\`{a}} Puigdom{\`{e}}nech Badia and
                  Mehdi Mirza and
                  Alex Graves and
                  Timothy P. Lillicrap and
                  Tim Harley and
                  David Silver and
                  Koray Kavukcuoglu},
  title        = {Asynchronous Methods for Deep Reinforcement Learning},
  journal      = {CoRR},
  volume       = {abs/1602.01783},
  year         = {2016},
  url          = {http://arxiv.org/abs/1602.01783},
  eprinttype    = {arXiv},
  eprint       = {1602.01783},
  timestamp    = {Mon, 13 Aug 2018 16:47:40 +0200},
  biburl       = {https://dblp.org/rec/journals/corr/MnihBMGLHSK16.bib},
  bibsource    = {dblp computer science bibliography, https://dblp.org}
}
@InProceedings{espeholt18,
  title = 	 {{IMPALA}: Scalable Distributed Deep-{RL} with Importance Weighted Actor-Learner Architectures},
  author =       {Espeholt, Lasse and Soyer, Hubert and Munos, Remi and Simonyan, Karen and Mnih, Vlad and Ward, Tom and Doron, Yotam and Firoiu, Vlad and Harley, Tim and Dunning, Iain and Legg, Shane and Kavukcuoglu, Koray},
  booktitle = 	 {Proceedings of the 35th International Conference on Machine Learning},
  pages = 	 {1407--1416},
  year = 	 {2018},
  editor = 	 {Dy, Jennifer and Krause, Andreas},
  volume = 	 {80},
  series = 	 {Proceedings of Machine Learning Research},
  month = 	 {10--15 Jul},
  publisher =    {PMLR},
  pdf = 	 {http://proceedings.mlr.press/v80/espeholt18a/espeholt18a.pdf},
  url = 	 {https://proceedings.mlr.press/v80/espeholt18a.html},
  abstract = 	 {In this work we aim to solve a large collection of tasks using a single reinforcement learning agent with a single set of parameters. A key challenge is to handle the increased amount of data and extended training time. We have developed a new distributed agent IMPALA (Importance Weighted Actor-Learner Architecture) that not only uses resources more efficiently in single-machine training but also scales to thousands of machines without sacrificing data efficiency or resource utilisation. We achieve stable learning at high throughput by combining decoupled acting and learning with a novel off-policy correction method called V-trace. We demonstrate the effectiveness of IMPALA for multi-task reinforcement learning on DMLab-30 (a set of 30 tasks from the DeepMind Lab environment (Beattie et al., 2016)) and Atari57 (all available Atari games in Arcade Learning Environment (Bellemare et al., 2013a)). Our results show that IMPALA is able to achieve better performance than previous agents with less data, and crucially exhibits positive transfer between tasks as a result of its multi-task approach.}
}

@article{vaswani2021,
  author       = {Sharan Vaswani and
                  Olivier Bachem and
                  Simone Totaro and
                  Robert Mueller and
                  Matthieu Geist and
                  Marlos C. Machado and
                  Pablo Samuel Castro and
                  Nicolas Le Roux},
  title        = {A functional mirror ascent view of policy gradient methods with function
                  approximation},
  journal      = {CoRR},
  volume       = {abs/2108.05828},
  year         = {2021},
  url          = {https://arxiv.org/abs/2108.05828},
  eprinttype    = {arXiv},
  eprint       = {2108.05828},
  timestamp    = {Wed, 18 Aug 2021 19:45:42 +0200},
  biburl       = {https://dblp.org/rec/journals/corr/abs-2108-05828.bib},
  bibsource    = {dblp computer science bibliography, https://dblp.org}
}
@article{agarwal2019,
  author       = {Alekh Agarwal and
                  Sham M. Kakade and
                  Jason D. Lee and
                  Gaurav Mahajan},
  title        = {Optimality and Approximation with Policy Gradient Methods in Markov
                  Decision Processes},
  journal      = {CoRR},
  volume       = {abs/1908.00261},
  year         = {2019},
  url          = {http://arxiv.org/abs/1908.00261},
  eprinttype    = {arXiv},
  eprint       = {1908.00261},
  timestamp    = {Fri, 09 Aug 2019 12:15:56 +0200},
  biburl       = {https://dblp.org/rec/journals/corr/abs-1908-00261.bib},
  bibsource    = {dblp computer science bibliography, https://dblp.org}
}
@article{shani2019,
  author       = {Lior Shani and
                  Yonathan Efroni and
                  Shie Mannor},
  title        = {Adaptive Trust Region Policy Optimization: Global Convergence and
                  Faster Rates for Regularized MDPs},
  journal      = {CoRR},
  volume       = {abs/1909.02769},
  year         = {2019},
  url          = {http://arxiv.org/abs/1909.02769},
  eprinttype    = {arXiv},
  eprint       = {1909.02769},
  timestamp    = {Mon, 16 Sep 2019 17:27:14 +0200},
  biburl       = {https://dblp.org/rec/journals/corr/abs-1909-02769.bib},
  bibsource    = {dblp computer science bibliography, https://dblp.org}
}
@article{abdolmaleki2018,
  author       = {Abbas Abdolmaleki and
                  Jost Tobias Springenberg and
                  Yuval Tassa and
                  R{\'{e}}mi Munos and
                  Nicolas Heess and
                  Martin A. Riedmiller},
  title        = {Maximum a Posteriori Policy Optimisation},
  journal      = {CoRR},
  volume       = {abs/1806.06920},
  year         = {2018},
  url          = {http://arxiv.org/abs/1806.06920},
  eprinttype    = {arXiv},
  eprint       = {1806.06920},
  timestamp    = {Mon, 13 Aug 2018 16:48:15 +0200},
  biburl       = {https://dblp.org/rec/journals/corr/abs-1806-06920.bib},
  bibsource    = {dblp computer science bibliography, https://dblp.org}
}
@inproceedings{sutton2000,
 author = {Sutton, Richard S and McAllester, David and Singh, Satinder and Mansour, Yishay},
 booktitle = {Advances in Neural Information Processing Systems},
 editor = {S. Solla and T. Leen and K. M\"{u}ller},
 pages = {},
 publisher = {MIT Press},
 title = {Policy Gradient Methods for Reinforcement Learning with Function Approximation},
 url = {https://proceedings.neurips.cc/paper_files/paper/1999/file/464d828b85b0bed98e80ade0a5c43b0f-Paper.pdf},
 volume = {12},
 year = {1999}
}

@misc{johnson2023,
      title={Optimal Convergence Rate for Exact Policy Mirror Descent in Discounted Markov Decision Processes},
      author={Emmeran Johnson and Ciara Pike-Burke and Patrick Rebeschini},
      year={2023},
      eprint={2302.11381},
      archivePrefix={arXiv},
      primaryClass={math.OC}
}
@misc{xiao2022,
      title={On the Convergence Rates of Policy Gradient Methods},
      author={Lin Xiao},
      year={2022},
      eprint={2201.07443},
      archivePrefix={arXiv},
      primaryClass={math.OC}
}
@article{tomar2020,
  author       = {Manan Tomar and
                  Lior Shani and
                  Yonathan Efroni and
                  Mohammad Ghavamzadeh},
  title        = {Mirror Descent Policy Optimization},
  journal      = {CoRR},
  volume       = {abs/2005.09814},
  year         = {2020},
  url          = {https://arxiv.org/abs/2005.09814},
  eprinttype    = {arXiv},
  eprint       = {2005.09814},
  timestamp    = {Fri, 22 May 2020 16:21:28 +0200},
  biburl       = {https://dblp.org/rec/journals/corr/abs-2005-09814.bib},
  bibsource    = {dblp computer science bibliography, https://dblp.org}
}
@misc{alfano2023,
	title        = {Linear Convergence for Natural Policy Gradient with Log-linear Policy Parametrization},
	author       = {Carlo Alfano and Patrick Rebeschini},
	year         = 2023,
	eprint       = {2209.15382},
	archiveprefix = {arXiv},
	primaryclass = {cs.LG}
}
@inproceedings{chen2022,
	title        = {Sample Complexity of Policy-Based Methods under Off-Policy Sampling and Linear Function Approximation},
	author       = {Chen, Zaiwei and Theja Maguluri, Siva},
	year         = 2022,
	month        = {28--30 Mar},
	booktitle    = {Proceedings of The 25th International Conference on Artificial Intelligence and Statistics},
	publisher    = {PMLR},
	series       = {Proceedings of Machine Learning Research},
	volume       = 151,
	pages        = {11195--11214},
	url          = {https://proceedings.mlr.press/v151/chen22i.html},
	editor       = {Camps-Valls, Gustau and Ruiz, Francisco J. R. and Valera, Isabel},
	pdf          = {https://proceedings.mlr.press/v151/chen22i/chen22i.pdf},
	abstract     = {In this work, we study policy-based methods for solving the reinforcement learning problem, where off-policy sampling and linear function approximation are employed for policy evaluation, and various policy update rules (including natural policy gradient) are considered for policy improvement. To solve the policy evaluation sub-problem in the presence of the deadly triad, we propose a generic algorithm framework of multi-step TD-learning with generalized importance sampling ratios, which includes two specific algorithms: the $\lambda$-averaged $Q$-trace and the two-sided $Q$-trace. The generic algorithm is single time-scale, has provable finite-sample guarantees, and overcomes the high variance issue in off-policy learning. As for the policy improvement, we provide a universal analysis that establishes geometric convergence of various policy update rules, which leads to an overall $\Tilde{\mathcal{O}}(\epsilon^{-2})$ sample complexity.}
}

**Questions:**

See above.

---

> ### Author Response · Authors · 2023-11-19
> **Response to reviewer kAky**
>
> Thank you for your feedback. We must clarify, however, that our paper does not address the issues of "0" and "true gradient" mentioned in your review. Moreover, we have not received specific questions regarding our manuscript. Should you have any particular concerns about our work, we earnestly request that you point out the key aspects of the Weaknesses and any questions. We would be honored to respond to each of your queries in detail. Lastly, we thank you for taking the time to engage with our work.

---

> > ### Comment · Reviewer_kAky · 2023-11-21
> > **Rebuttal response**
> >
> > The other reviews and responses helped me understand the paper somewhat better.
> >
> > However I'm still not completely sure I understand the motivation. Is the point to transform the d_{pi_t} stationary distribution w.r.t. to the previous policy pi_t back into an on-policy distribution, or re-weight the trajectory data in each iteration?

---

> > > ### Author Response · Authors · 2023-11-21
> > > **Response to the follow-up questions**
> > >
> > > Thank you for your feedback. We greatly appreciate your constructive suggestions, which have provided valuable guidance for enhancing the quality of our manuscript. We will try to answer your questions as best we can. Lastly, we thank you for taking the time to engage with our work.
> > >
> > > **Question 1**: The motivation for this paper.
> > >
> > > **Answer 1**: Existing algorithms optimize the policy by the value function and don’t **directly** consider utilizing subsequent data to optimize the current policy, which may be a reason for sample inefficiency.
> > >
> > > Intuitively, the direct use of the subsequent data may speed up the convergence of the algorithm and improve sample efficiency. In the subsequent sections of this paper, we verify this intuition theoretically and experimentally, respectively.
> > >
> > >
> > >
> > > **Question 2**: Is the point to transform the $d_{\pi_t}$ stationary distribution w.r.t. to the previous policy $\pi_t$ back into an on-policy distribution, or re-weight the trajectory data in each iteration?
> > >
> > > **Answer 2**: The point is to transform the $d_{\pi_t}$ stationary distribution into on-policy distribution. Below, we give an explanation from another perspective.
> > >
> > > For the surrogate function of TRPO, defined
> > >
> > > $ \hat{L}(\pi, \pi_t)=E_{s,a\sim d_{\pi_t}(\cdot)} r A^{\pi_t}(s, a)$
> > >
> > > We show intuitively how to optimize the upper formula. We sample ($s,a$) to optimize the above equation. The state $s$ is viewed as sampled from the stationary distribution $d_{\pi_t}(s)$. Assuming that we sample ($s,s'$) from the environment (omit action $a$) and that $s'$ is the next state of $s$. Similarly, The state pair ($s,s'$) is viewed as sampled from the stationary distribution $d_{\pi_t}(s, s')$ (defined similarly to $d_{\pi_t}(s)$). The conditional distribution of $s'$ under the distribution of $s$ can be calculated as $\frac{d_{\pi_t}(s,s')}{d_{\pi_t}(s)}=d_{\pi_t}(s'|s)$.
> > >
> > > In a practical environment, ($s,s'$) is approximated as obtained by sampling from the above distribution. We optimize Eqn. (6) by this way.

---

> > > ### Author Response · Authors · 2023-11-21
> > > **Summarise the issues and respond to reviewer kAky**
> > >
> > > Thank you for your feedback. Since you haven't summarized the issue, we have summarized it. We will do our best to answer your questions. Lastly, we thank you for taking the time to engage with our work.
> > >
> > > **Question 1**: Explanations of some text.
> > >
> > > **Answer 1**: I'm very sorry to have caused you some confusion. I will give as detailed an explanation as possible below: 1) From the theorem 4.1, we establish a relationship on the performance of policies $\pi$ and $\hat{\pi}$, which is a theoretical extension of TRPO, as well as giving a new way of using the subsequent data to directly assist in optimizing policy and adding some insights to the policy gradient algorithm. From Theorem 4.2, we give an advantage of the proposed method, *i.e.*, the ability to reduce the solution space, which can find a good policy more efficiently. Therefore, we say “We give a nice theory”; 2) About “RPO efficiently utilizes ‘good’ experiences, makes adjustments based on ‘bad’ experiences" and "the policy can be reflective". We did a visualization experiment on the CliffWalking environment. Fig. 2 shows that RPO significantly reduces the frequency of falling off the 'Cliff' and can reach the target 'G' faster than PPO  under equal iteration conditions, which further confirms the benefits of utilizing the subsequent data. The successful implementation of RPO is attributed to directly using subsequent states to adjust the actions of the current state, which suggests that the policy can be reflective.
> > >
> > > **Question 2**: About “this term “1” may adversely affect policy optimization" and “for $k=1$, the $l_1$ norm constraints are replaced by KL constraints”.
> > >
> > > **Answer 2**: Consider $L_1(\pi, \hat{\pi})$ in Eqn. (2) as an example. If the environment is unknown, it can only be optimized by sampling. If we sample a trajectory $\tau$, and $(s_0, a_0, s_1, a_1)\in\tau$, if $ A^{\hat{\pi}}(s_1, a_1)<0 $ and $ r_0-1 <0$, we know that $ (r_0-1)r_1 A^{\hat{\pi}}(s_1, a_1)>0 $. Optimizing the $ L_1(\pi, \hat{\pi}) $ (considering the extreme case, using a sample), the probability of $ a_1 $ is increased. However, $ A^{\hat{\pi}}(s_1, a_1)<0 $, we should  decrease the probability of $ a_1 $. It's a contradiction. Thus this term "1" of $ r_0-1 $ may adversely affect policy optimization though the theory is sound. This situation exists when the environment is unknown. About "the $l_1$ norm constraints are replaced by KL constraints", this is to increase the connection between the paper and previous research and to better present the completion of the paper.
> > >
> > > **Question 3**: About the importance sampling ratio "$r$"and “$k$”.
> > >
> > > **Answer 3**: In this paper, we define $r_t=\frac{\pi(a_t|s_t)}{\hat{\pi}(a_t|s_t)}$ of Eqns. (2-3), which denotes importance sampling at step $ t $. When $k=1$, this means that that agent performs an action to move from one visitation distribution to another visitation distribution, *i.e*., the agent performs action $\hat{\pi}(a_0|s_0)$ from state $ s_0 $ ($ s_0\in\rho^{\hat{\pi}}(\cdot) $ is denoted as $ s_0 $ is a random variable of the discount state visitation distribution $\rho^{\hat{\pi}}(\cdot) $) to the next state $ s_1 $ ($ s_1\in\rho^{\hat{\pi}}(\cdot|s_0, a_0) $ is denoted as $ s_1 $ is a random variable of the conditional discount state visitation distribution $\rho^{\hat{\pi}}(\cdot|s_0, a_0) $). Similarly, for any $k$, the agent performs $k$ action move from $ s_0 $ ($ s_0\in\rho^{\hat{\pi}}(\cdot)$) to state $ s_k $ ($ s_k\in\rho^{\hat{\pi}}(\cdot|s_{k-1}, a_{k-1}) $). Hence, we give the difference between the performance of $ \hat{\pi} $ and $ \pi $ after $ k $ times from the theorem 3.1 and 4.1. When $ k =1 $, our result is consistent with TRPO.
> > >
> > > **Question 4**: The authors claim the method maintains monotonic improvement, but it is known that proximal-gradient methods, with adaptive step size have this property.
> > >
> > > **Answer 4**: We have carefully read the paper you have provided. Let me elaborate on the differences between our method and the above method below. The studies you mentioned only consider the current state value function to optimize the policy, this paper considers not only the current state but more importantly the direct effect of subsequent data on the current to optimize policy, as in theorem 3.1 and 4.1. So, the theorems obtained from previous studies do not apply to the method proposed in this paper and we give a new interesting theory.

---

> > > > ### Comment · Reviewer_kAky · 2023-11-22
> > > > **Re**
> > > >
> > > > Q1: not sure this definition is very specific, mixing Monte-Carlo with TD would be using subsequent experience, but I don't think that is what you are doing, is it? Whether the value function is insufficient should depend on its accuracy, no?
> > > > Once the objective is linearized and a lower bound surrogate objective function defined, optimizing that to optimality guarantees improvement, and at optimal rate under a specific coefficient/step-size/proximal regularization in front of the KL.
> > > > If your value function is inaccurate by some epsilon, the sample efficiency quantifies how many calls you need to obtain an epsilon'-accurate policy.
> > > >
> > > > Q2 & Q3: 'r' is generally used for the reward in RL, I think got confused a few times because I need to reload this conflicting notation again in memory. Why not use any other letter/symbol in the alphabet(s)?
> > > >
> > > > Q2 A linear model of the policy performance is not a lower bound no? Nor is there anywhere any claim of such right? Only a quasi-Newton model can be a lower bound under a specific regularization. Not sure I understand this example?
> > > >
> > > > Q2 & Q3: not sure I understand why you are considering L1 norms and you move between Bregman divergences and L1 norms without careful explanation of these steps
> > > >
> > > > Q4 The adaptive step-size proximal method converges at the gamma rate of policy iteration, which is unimprovable. For sample efficiency, there should be a difference in the number of calls to the environment between a method that uses an eps-inaccurate value function and your version, which I understand improves by using the same data. I assume the difference comes from using the same data between in absence of a model. I would like to understand this point better. Is that where your main contribution is?

---

> ### Author Response · Authors · 2023-11-23
> **Response to Re**
>
> Thank you for your feedback. I'm sorry I don't particularly understand what you're saying. We will do our best to answer your questions. Lastly, we thank you for taking the time to engage with our work.
>
> **Re-Question 1**: Did you do that mixing Monte-Carlo with TD would be using subsequent experience? Accuracy of the value function?
>
> **Re-Answer 1**: No, my proposed method doesn't use it. Our approach is in line with TRPO [1] and CPO [2]. The key of those method is to not update the policy too much, thus making the algorithm more stable, when the dynamic model is unknown.
>
> Below, we we illustrate the differences between our proposed approach RPO and TRPO [1].
>
> The objective function of TRPO [1] (derived rather than adding regular terms):
>
> $$E_{s,a}[\frac{\pi(a|s)}{\hat{\pi}(a|s)} A^{\hat{\pi}}(s, a)]- \hat{C}_{1}(\pi, \hat{\pi}),$$
>
> where  $\hat{C}_{1}(\pi, \hat{\pi})=\frac{\gamma R\max}{(1-\gamma)^2}\|\pi-\hat{\pi}\|_1^2$.
>
> When $k=2$, the objective function of my method RPO (derived rather than adding regular terms):
>
> $$E_{s,a}[\frac{\pi(a|s)}{\hat{\pi}(a|s)} A^{\hat{\pi}}(s, a)]+\alpha E_{s,a, s',a'}[\frac{\pi(a|s)}{\hat{\pi}(a|s)}\frac{\pi(a'|s')}{\hat{\pi}(a'|s')} A^{\hat{\pi}}(s', a')] - \hat{C}_{2}(\pi, \hat{\pi}),$$
>
> where $\alpha$ is a constant.
>
> Our approach adds an additional  term compared to TRPO. It can be seen that our method is very different from previous methods [1,2,3]. The methodology of previous studies [3,4,5] of sample efficiency does not directly apply in my case.
>
> Note that the objective function of TRPO can be obtained in terms of adding the regular term (Bregman divergences, *e.g.*, KL), but the original objective functions of TRPO is derived from policy performance (see paper TRPO [1] and my paper).
>
> **Re-Question 2**: About 'r' and example.
>
> **Re-Answer 2**: Thank you very much for your suggestion. We will use a different symbol to represent it.
>
> We approach it from an optimization perspective, considering the action probabilities of the policy, and the penalty term prevents the policy updates from being too large. In the TayPO paper [6], they use $L_0(\pi, \hat{\pi})+L_1(\pi, \hat{\pi})$ as an objective function and omit the penalty term. Optimizing this objective function may pose issues. We consider this function without focusing on the specific form of the parameters. When the environment is unknown, it can only be optimized by sampling. Considering the extreme case, the function  $L_1(\pi, \hat{\pi})$ is optimized by using a sample $(s_0, a_0, s_1, a_1)$, *i.e.*, $L_1(\pi, \hat{\pi})\approx(I_0-1)I_1 A^{\hat{\pi}}(s_1, a_1) $.
>
> If $ A^{\hat{\pi}}(s_1, a_1)<0 $ and $ I_0-1 <0$, we know that $ (I_0-1)I_1 A^{\hat{\pi}}(s_1, a_1)=[(I_0-1)A^{\hat{\pi}}(s_1, a_1)]I_1>0 $. The probability of $ a_1 $ is increased. However, when $ A^{\hat{\pi}}(s_1, a_1)<0 $, we should  decrease the probability of $ a_1 $. It's a contradiction. Thus, this term "1" of $ I_0-1 $ may adversely affect policy optimization, though the theory is sound. This situation exists when the environment is unknown.
>
> **Re-Question 4**: About "sample efficiency" and "contribution ".
>
> **Re-Answer 4**: I'm guessing you're asking both questions.
>
> Sample efficiency refers to how many samples are needed to achieve the same level of performance. In other words, how much algorithm performance can be achieved using the same samples.
>
> From theorem 4.2, we shows that the solution space is contracting when $k=2$. Reducing the solution space is beneficial in accelerating the convergence of the algorithm in some sense. Better performance can be achieved in the same number of steps of the algorithm iteration. In other words, with the same number of samples, we achieve better policy performance.
>
> We generalized the TRPO algorithm, and proposed a new lower bound, combining the current data and subsequent data to optimize the current policy. Theoretical analyses show that our proposed method, in addition to satisfying the monotonic improvement of policy performance, can effectively reduce the solution space of the optimized policy, resulting in speeding up the training procedure of the algorithm.
>
>
>
>
>
> [1] Schulman J, Levine S, Moritz P,et al. Trust Region Policy Optimization. ICML, 2015.
>
> [2] Achiam J , Held D, Tamar A,et al. Constrained Policy Optimization. ICML, 2017.
>
> [3] Hepeng L and Nicholas C ,et al. An Analytical Update Rule for General Policy Optimization. ICML, 2022.
>
> [4] Jalaj B and Daniel R. On the Linear Convergence of Policy Gradient Methods for Finite MDPs. AISTATS, 2021.
>
> [5] Alekh A and Sham M,et al. Optimality and Approximation with Policy Gradient Methods in Markov Decision Processes. 2020.
>
> [6] Yunhao T,and Michal V,et al. Taylor expansion policy optimization. ICML, 2020.

---

### Meta-Review · Area_Chair_2DfF · 2023-12-05

**Metareview:**

This work extends the policy optimization surrogate from one step to multiple steps, akin to [1]. The final error term is proved to be smaller than [1]. Empirical study demonstrates some performance improvement over existing methods.

Strengths: This work provides a new way for extending one step policy optimization to multi steps. Some performance improvements are observed empirically.

Weakness: Though I in general appreciate the idea of this work, the current presentation significantly hurts its accessibility to the reader and lacks rigour (see kAky's comment). The theoretical results could also benefit from a better presentation. Moreover, the relationship between this work and [1] is not sufficiently addressed. Given the similarity to [1], I would suggest the authors rephrase this work as a technical improvement over [1] based on the Corollary A.3. Currently, this work is motivated from a reflection perspective, which is a very subjective word and lacks technical rigor. As seen by multiple reviewers, this motivation is not convincing and is a major source of the confusion. One important question the authors need to address is that what the difference is between this work and a simple clipping version of [1] and how the tighter bound in Corollary A.3 translates into (possibly) better performance than [1].

[1] Tang, Yunhao, Michal Valko, and Rémi Munos. "Taylor expansion policy optimization." International Conference on Machine Learning. PMLR, 2020.

**Justification For Why Not Higher Score:**

The current presentation of this work is very confusing and this work lacks sufficient comparison with [1].

**Justification For Why Not Lower Score:**

N/A

---

### Decision · Program_Chairs · 2024-01-16

Reject